# ONE CLUSTER OR TWO? A MANIFOLD-BASED APPROACH

## ABSTRACT

The manifold hypothesis suggests a natural criterion for clustering: partition data according to the manifold component from which each point is drawn. This criterion is useful because, intuitively, the separability of manifold components depends on how their ambient separation between components compares to the largest sampling gap. The analysis integrates topology (e.g., manifold volume and reach) with estimation (e.g., fill radius and sample density). Formally, it identifies a criticality: when a threshold is exceeded, nearest-neighbor data graphs avoid bridging edges and clusters are preserved; otherwise, bridges appear and components fuse. In practice, this critical threshold lies between bounds that imply a measure of cluster confidence and motivates an algorithm—Manifold-Based Clustering (MBC)—that constructs a candidate neighborhood graph. MBC is parameter-light and, unlike density-based methods (e.g., HDBSCAN), avoids hand-tuned scale thresholds. Instead, MBC yields a monotone bound, or *bracket*, on the number of components by a natural sweep of neighborhood size. Across curved and high-dimensional benchmarks, MBC matches state-of-the-art accuracy and exposes ambiguity near the critical thresholds.

## 1 INTRODUCTION

Clustering is a notoriously thorny problem. Results depend on criteria (Kleinberg, 2002), separation (Hennig, 2015), and sampling (Tibshirani et al., 2001), among other factors. To address this, researchers traditionally rely on domain knowledge (e.g. genomics (Eisen et al., 1998)) or on popular algorithms (McInnes et al., 2018; Ester et al., 1996; Ankerst et al., 1999; Campello et al., 2013; 2015). However, the statistical power of these algorithms is difficult to assess (Dalmaijer et al., 2022), and blindly using any of them could be problematic (Chari & Pachter, 2023). This is especially true in neuroscience (Button et al., 2013), where even determining whether data are (in fact) clustered remains an important open problem (Dyballa et al., 2024b).

We illustrate two extremes in Fig. 1. Typically, when one mentions clusters, an artificial image comes to mind as shown in blue and orange: two collections of well-separated data points. However, in reality, data can be distributed as the neuroscience plots at the top. These data are drawn from recordings of retinal ganglion cells, and the neuroscientists involved estimate that there are about eight clusters of cells, based on separately measured physiological properties (Dyballa et al., 2024a)). Applying the above traditions, popular algorithms are off by a factor of 3–4. We embrace the uncertainty directly, and propose an algorithm that yields a bracket [1–9] for the number of clusters. The algorithm is based on topological analysis, and takes both shape and sampling into consideration. Thus, clusters may be clearly separable, non-separable, or lie in a transitional regime. The neural data falls within the latter two.

To motivate the analysis, we ask: *were the data sampled from a connected or separated object, and by what margin?* Adopting the manifold hypothesis, we model high-dimensional observations as samples from a compact subset $\mathcal{M} \subset \mathbb{R}^D$ that is either a single connected $C^2$ submanifold or a finite union of disjoint $C^2$ components (Fefferman et al., 2023). This viewpoint reframes clustering as a *decision problem*: given i.i.d. samples $X = \{x_i\}_{i=1}^n$ from a distribution supported on $\mathcal{M}$, can it be decided whether the support is connected or decomposes into separated components? A threshold criterion emerges that determines whether clusters exist and, by extension, estimates their number. The resulting bracket captures this threshold region.

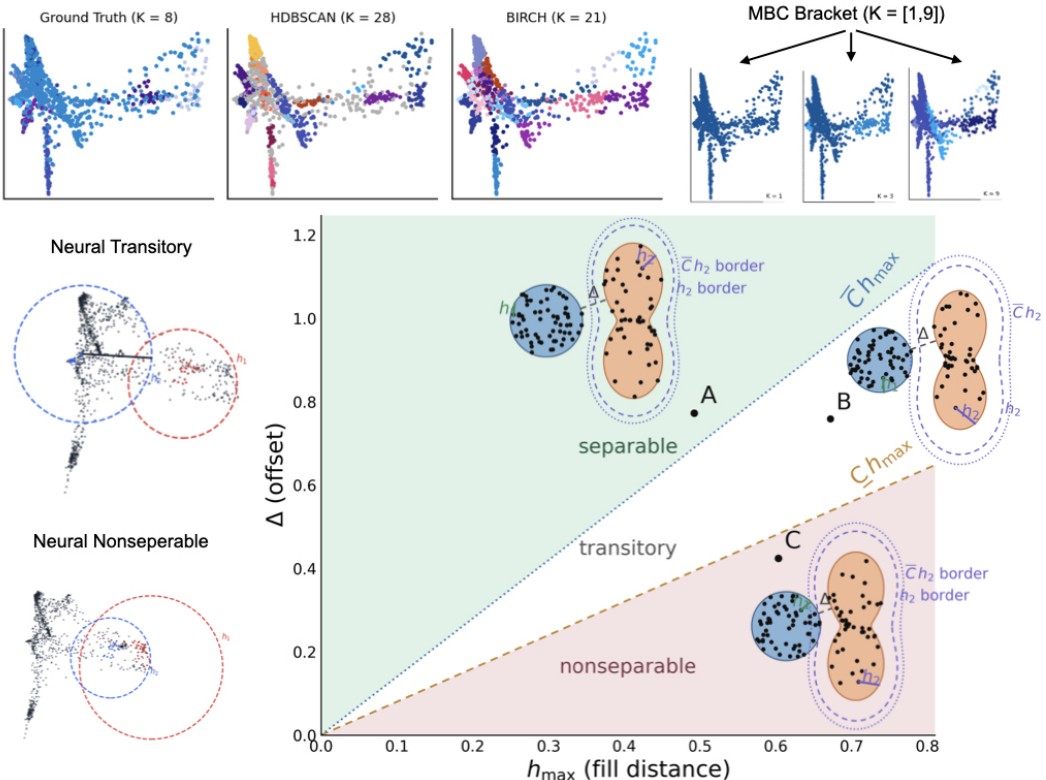

Figure 1: **Motivating example.** Neural spiking data are often mixed, with clusters embedded in significant noise. We show how the number of clusters, estimated on physiological grounds to be about 8, cannot be reliably determined by standard unsupervised clustering algorithms (HDBSCAN and BIRCH). The MBC algorithm provides a feasible bracket for the number of clusters that includes the physiologically expected number. We highlight the geometric principle of cluster separation in the main panel, where **(A–C)** depict a non-pure manifold with two components: a disc (left, in blue) and a softened peanut (right, in orange). We show worst-case fill distances ($h_1, h_2$), the minimal offset $\Delta$, and two boundaries of the peanut, $h_2$ (dashed) and $\overline{C} h_2$ (dotted) where $\overline{C}$ is a constant determined by the geometry of the manifold. The ratio $\rho = \Delta/h_{\max}$, where $h_{\max} = \max\{h_1, h_2\}$, governs separability: **A** is separable ($\rho > \overline{C}$), **B** is transitional ($\underline{C} < \rho < \overline{C}$), and **C** is non-separable ($\rho < \underline{C}$, with $\Delta > 0$). We also provide examples of transitional and non-separable true neural clusters.

The criterion that governs this decision is estimated from $k$-nearest neighbor ($k$NN) graphs: the ratio $\rho$ between the *offset* $\Delta$ (the minimal Euclidean distance between any two components) and the *fill distance* $h$ of the sample over those components (the worst-case sampling gap). Intuitively, the fill distance measures the size of the largest hole in the sample coverage; smaller fill distance implies denser, more uniform sampling. Thus, large values of $\rho$ indicate the presence of clearly separated clusters (relative to sample density), while small values mean the estimated components should be blurred together into a single cluster. Classic random geometric graph (RGG) results justify this strategy: RGGs exhibit sharp connectivity thresholds as the neighborhood scale changes with sample size $n$. They become connected around radii $r_n \asymp ((\log n)/n)^{1/d}$ or when the $k$-NN parameter scales like $k \asymp \log n$, under mild regularity conditions (Penrose, 2003; Balister et al., 2005). We translate this picture to the problem of separating manifold components. In our setting, the constants depend only on standard intrinsic geometry properties such as the two-sided volume growth (bounds small-ball volumes) and positive reach (Niyogi et al., 2008). This translation allows us to formally quantify when distinct manifold components will remain disconnected in a $k$NN graph, rather than linked by spurious 'bridging' edges.

We leverage this result into an algorithm (MBC), extend it to the tubular noise regime, and compute the confidence bracket using a fill-distance approximation, resulting in the following contributions:

1. **Geometric criterion for cluster preservation.** We introduce the offset-fill-distance ratio and prove upper and lower thresholds that predict when clusters remain distinct in the standard and noisy regimes (Theorem 3.3).

2. **Manifold-Based Clustering Algorithm (MBC).** We develop an algorithm that applies this threshold to uncover clusters and handle noise robustly using the distance-to-measure framework (Algorithm 2).

3. **Density criterion for bracket relaxation.** We exploit the derivation of MBC to develop Corollary 4.1, reflecting the uncertainty in the underlying number of clusters with respect to the framework established in Theorem 3.3.

Finally, we show empirically that the bracket captures $k$, the number of clusters, for synthetic datasets as well as the neuroscience example shown in Fig. 1. Blindly applying popular algorithms can be misleading; instead, we highlight the uncertainty in clustering real-world data, and the delicate interplay between sampling and topology inherent in this task.

## 2 BACKGROUND

*Neighborhood graphs and threshold scales.* Manifold-learning methods—Isomap, LLE, Laplacian Eigenmaps—reconstruct geometry from neighborhood graphs using shortest-path or spectral surrogates (Tenenbaum et al., 2000; Roweis & Saul, 2000; Belkin & Niyogi, 2003; Coifman & Lafon, 2006), and spectral clustering relies critically on the quality of the same neighborhood graph (von Luxburg, 2007; Zelnik-manor & Perona, 2004). Popular dimensionality reduction methods such as UMAP explicitly optimize objectives that preserve local neighborhoods (McInnes et al., 2018). The reliability of these pipelines depends on choosing neighborhoods that match the intrinsic sampling scale: if neighborhoods are too large, graphs connect across gaps and destroy component structure. Random graph theory formalizes this with sharp transitions: connectivity emerges at radii $r_n \asymp (\log n/n)^{1/d}$, and union-$k$NN graphs become connected when $k \asymp \log n$, with constants depending on dimension and local volume regularity (Penrose, 2003; Balister et al., 2005). We leverage these scales in practice by setting $k = O(\log n)$ so the graph lies near its connectivity threshold—neither too sparse (disconnected) nor too dense (over-connected). Moreover, we 'bracket' the true number of meaningful components in the data between two close values for $k$, thus defining a *confidence bracket* in a loose statistical sense.

*Fill distance, two-sided volume growth, and uniform $k$NN radii.* The fill distance $h(R, \mathcal{M}) = \sup_{x \in \mathcal{M}} \min_{r_i \in R} \|x - r_i\|$ is the worst-case sampling gap on $\mathcal{M}$. Under two-sided volume growth (i.e., lower and upper bounds on small-ball volumes) and positive reach, covering radii and nearest-neighbor distances concentrate uniformly around the intrinsic sampling scale; in particular, for samples on a $d$-dimensional support, $h$ and $k$NN radii $D_k(x)$ scale respectively like $((\log n)/n)^{1/d}$ and $(k/n)^{1/d}$ up to constants (Niyogi et al., 2008; Boissonnat et al., 2018). Our separability condition compares $\Delta$ to $h_{\max}$ across components; when $\Delta/h_{\max}$ exceeds a curvature-dependent constant, stabilized $k$NN neighborhoods do not mix components.

*Relationship to reach and curvature.* The reach $\tau_M$ of a smooth subset $M \subset \mathbb{R}^D$ is the largest radius for which every point in the tubular neighborhood of $M$ has a unique nearest-point projection onto $M$ (Federer, 1959); equivalently, it is the infimum distance from $M$ to its medial axis, i.e. the set of points with multiple nearest neighbors. Reach captures both local curvature—$\tau_M$ is bounded above by the reciprocal of the largest principal curvature—and global bottlenecks, since narrow necks shrink $\tau_M$. Practical estimators recover $\tau_M$ and related geometric quantities from point samples, with non-asymptotic guarantees (Aamari et al., 2019); recent analyses clarify how reach behaves for unions and under set operations (Boissonnat & Wintraecken, 2023). The ratio $\Delta/h$ can be interpreted as a relaxation of the reach, tailored to *distinct components*: $\Delta$ is twice the bottleneck radius *between* components in the medial-axis picture, while $h$ measures sample dispersion. Requiring $\Delta/h$ to exceed a constant ensures that sampling density lies below the relevant bottleneck scale, preventing spurious graph connections between components.

*Robust local statistics, transitivity, and density-based clustering.* Raw Euclidean distances are notoriously sensitive to density variation and moderate noise. The distance-to-measure (DTM), which averages local nearest-neighbor distances, provides a robust and scale-aware alternative with stability guarantees (Chazal et al., 2011). A directional two-scale DTM cancels leading density bias on the

manifold, yet grows linearly with ambient offset; this property underpins our conservative add-only rescue procedure. Requiring shared-neighbor (triangle) support suppresses spurious asymmetric short links and enforces minimal transitivity, cf. shared-nearest-neighbor clustering (Jarvis & Patrick, 1973). Density-based methods such as DBSCAN, BIRCH, OPTICS, and HDBSCAN infer clusters by thresholding density or mutual-reachability graphs and rely on user-specified parameters that implicitly decide whether bridges persist (Ester et al., 1996; Ankerst et al., 1999; Campello et al., 2013; 2015; Zhang et al., 1996). In contrast, our approach grounds this decision on a geometric offset–sampling scale, eliminates the need to hand-tune bridging thresholds, and yields a monotonic bracket on the component count by varying $k$ within a principled confidence range.

## 3    GEOMETRIC CLUSTER-SEPARATION CRITERION

We now introduce our main theoretical framework for understanding cluster separability in the manifold setting. Drawing parallels to Gaussian Mixture Models, we regard *offset* as the analogue of inter-cluster distance and *fill distance* as a proxy for "variance" or dispersion within each manifold component. We next extend this framework into an algorithm for detecting clusters under uncertainty. The procedure motivated by our framework is outlined in Figure 2.

Suppose our data lie on the union

$$\mathcal{M} \; = \; \mathcal{M}_1 \cup \cdots \cup \mathcal{M}_K, \; \mathcal{M}_i \cap \mathcal{M}_j = \emptyset, \; \forall i \neq j \in \{1, \ldots, K\}$$

where each $\mathcal{M}_k$ is a connected manifold component in $\mathbb{R}^D$. Let

$$\Delta \; = \; \min_{k \neq \ell} \Big\{ \|x - y\| : x \in \mathcal{M}_k, \, y \in \mathcal{M}_\ell \Big\}$$

be the *offset* (minimal ambient distance) between distinct components. In parallel, define the fill distance for $\mathcal{M}$'s sampled approximation as follows:

**Definition 3.1** (Fill Distance). Let $R = \{r_i\}_{i=1}^n \subset \mathcal{M}$ be a finite point set. The *fill distance* is

$$h_{R,\mathcal{M}} \; = \; \sup_{x \in \mathcal{M}} \; \min_{1 \leq i \leq n} \|x - r_i\|.$$

We say $R$ is quasi-uniform if $h_{R,\mathcal{M}}$ and the minimum pairwise distance among $r_i, r_j$ differ only by a constant factor. A smaller fill distance indicates that $R$ provides a denser covering of $\mathcal{M}$.

*Remark* 3.2. In analogy to the sampling density criterion and variance in Gaussian Mixture Models, we treat fill distance $h_{R,\mathcal{M}}$ as a measure of sampling dispersion. A smaller $h_{R,\mathcal{M}}$ corresponds to higher sampling density, which is often necessary for manifold learning algorithms to reliably approximate geodesic distances and local neighborhoods.

We denote $h_{R,\mathcal{M}}$ as $h$ for convenience and then consider the following ratio: $\rho \; = \; \frac{\Delta}{h}$.

### 3.1    MANIFOLD SEPARATION CRITERION

We now establish a threshold phenomenon governing the connectivity of $k$NN graphs constructed on points sampled from two disjoint, compact, $d$-dimensional Riemannian manifolds. We prove that in a $k$NN graph there exists a sharp transition, or threshold: when manifolds are far enough apart relative to sampling density, no edges cross; when they are close enough, bridging edges appear with high probability. In other words, under the assumption that clusters are separate if and only if they are sampled from two distinct manifold components, this theorem quantifies how sampling density (as measured by the fill distance) and intrinsic separation together determine whether the components remain disconnected or become connected in the $k$NN graph.

**Theorem 3.3** (Threshold for manifold separation in the union-$k$NN graph). *Let $\mathcal{M}_1, \mathcal{M}_2 \subset \mathbb{R}^D$ be disjoint, compact, connected, $d$-dimensional $C^2$ submanifolds with positive reach. Assume there exist constants $0 < \underline{c} \leq \overline{c} < \infty$ and a radius $r_* > 0$ such that for all $x \in \mathcal{M}_i$ and $0 < r \leq r_*$,*

$$\underline{c} \, r^d \; \leq \; \mu_i\big(B(x,r)\big) \; \leq \; \overline{c} \, r^d.$$

*Here $\mu_i$ denotes the normalized surface measure on $\mathcal{M}_i$.*

*Sampling. Independently draw $n_1$ and $n_2$ samples from $\mu_1$ and $\mu_2$, respectively; write $n = n_1 + n_2$ and $n_{\min} = \min\{n_1, n_2\}$. Let $S_i$ denote the set of sampled points on $\mathcal{M}_i$, define the fill distances*

$$h_i \;=\; \sup_{x \in \mathcal{M}_i} \min_{z \in S_i} \|x - z\|, \qquad h_{\max} \;=\; \max\{h_1, h_2\},$$

*and the ambient offset $\Delta \;=\; \inf\{\|x - y\| : x \in \mathcal{M}_1, \, y \in \mathcal{M}_2\}$.*

*Graph construction. Form the* union-$k$NN *(symmetrized $k$NN) graph using $k \;=\; \lceil A \log(4n/\delta) \rceil$, where $\varepsilon \in (0, 1)$ is fixed and $A \geq 3/\varepsilon^2$.*

***Threshold statement.*** *There exist explicit constants $\overline{C}, \underline{C} > 0$ (depending only on $d$, $\underline{c}$, $\overline{c}$, $A$, and $\varepsilon$) such that, with probability at least $1 - \delta$, the following hold:*

  *(i) If $\Delta/h_{\max} > \overline{C}$, then no edges connect $\mathcal{M}_1$ and $\mathcal{M}_2$.*

  *(ii) If $\Delta/h_{\max} < \underline{C}$ and $B\Delta \leq r_*$ for some $a \in (0, 1/8)$ with $B = 1 + 2a$, then the graph contains a cross edge with probability at least*

$$1 \;-\; 2\exp\big(-\underline{c}\, a^d\, n_{\min}\Delta^d\big) \;-\; \exp(-\gamma k),$$

  *for a universal constant $\gamma > 0$.*

*Remark 3.4* (Scaling of the thresholds). Let $R = \log(4n/\delta)\big/\log(n_{\min}/\delta)$ and $M = (\overline{c}/\underline{c})^{1/d}$. Then

$$\overline{C} \;=\; \Theta\big(A^{1/d}M\,R^{1/d}\big), \qquad \underline{C} \;=\; \Theta\big(A^{1/d}/(BM)\big).$$

In particular, under balanced sampling ($R \approx 1$), fixed $\varepsilon$ and $a$, and bounded geometry ($\overline{c}/\underline{c} = \Theta(1)$), both thresholds are $\Theta(A^{1/d})$, with constants depending only on $d$.

**Proof sketch.** The fill distances satisfy $h_i \asymp (\log(n_i/\delta)/n_i)^{1/d}$ with explicit upper and lower constants from a standard covering/packing argument under the local mass bounds, hence $h_{\max} \geq C_{\mathrm{fill}} (\log(n_{\min}/\delta)/n_{\min})^{1/d}$. Choosing $k = \lceil A \log(4n/\delta) \rceil$ and applying Chernoff with a union bound over all $n$ sample locations yields a uniform upper bound on the $k$NN-radius $D_k(Z) \leq (1-\varepsilon)^{-1/d}\big(2k/(n_{\min}\underline{c})\big)^{1/d}$, for every sample $Z$. Dividing by the lower fill bound yields $D_k(Z) \leq \overline{C}\, h_{\max}$ with $\overline{C}$ as above, so if $\Delta > \overline{C}\, h_{\max}$, no cross-edge is possible. For bridging, fix $a \in (0, 1/8)$ and $B = 1 + 2a$ and assume $B\Delta \leq r_*$. Occupancy of intrinsic caps of radius $a\Delta$ on each manifold occurs with probability at least $1 - 2\exp(-\underline{c}\, a^d\, n_{\min}\Delta^d)$, and an upper-mass Chernoff bound ensures that within radius $B\Delta$ around the near-boundary sample there are fewer than $k$ same-component neighbors with probability at least $1 - \exp(-\gamma k)$ provided $\Delta/h_{\max} < \underline{C}$. In that event, the cross-manifold sample lies within distance $B\Delta$ and must enter the top-$k$, producing a bridging edge. We offer full details, along with extensions to gaussian kernel graphs, in Appendix A.2.

## 3.2 EXTENDING CRITERION TO NOISY REGIMES

Empirical samples rarely lie exactly on a smooth manifold; instead, one observes noise as a tubular perturbation. This may shrink the separation between components and inflate the neighborhood radii. To account for this, we replace the original offset $\Delta$ by an effective offset $\Delta_{\mathrm{eff}}$, and show that $k$NN radii remain well-behaved. We adopt the following model: each component $\mathcal{M}_s \subset \mathbb{R}^D$ is compact, connected, $C^2$, with reach $\tau_s > 0$, and data points are of the form $x = \pi_{\mathcal{M}_s}(x) + \xi$, where $\pi_{\mathcal{M}_s}$ denotes nearest-point projection (well-defined whenever $\|\xi\| < \tau_s$) and $\xi$ is a mean-zero ambient perturbation that is either bounded almost surely by $\sigma < \tau_{\min} := \min_s \tau_s$ or sub-Gaussian with scale $\sigma$. In this regime the relevant offset becomes an effective quantity $\Delta_{\mathrm{eff}}$ satisfying $\Delta - 2\sigma \leq \Delta_{\mathrm{eff}} \leq \Delta + 2\sigma$ with high probability, while $k$NN radii concentrate around their noiseless counterparts with an additive $O(\sigma)$ deviation when $k \asymp \log(n/\delta)$.

The next statement upgrades the noiseless radius control used in Theorem 3.3 to the tubular-noise model and will allow us to distinguish between connected and separated components.

**Proposition 3.5** (Uniform $k$NN radii under tubular noise)**.** *Under the assumptions above, with probability at least $1 - \delta$, for every sample $x$ drawn from component $\mathcal{M}_s$,*

$$\underline{C}_s\, h_s \;-\; C_1\sigma \;\leq\; D_k(x) \;\leq\; \overline{C}_s\, h_s \;+\; C_2\,\sigma,$$

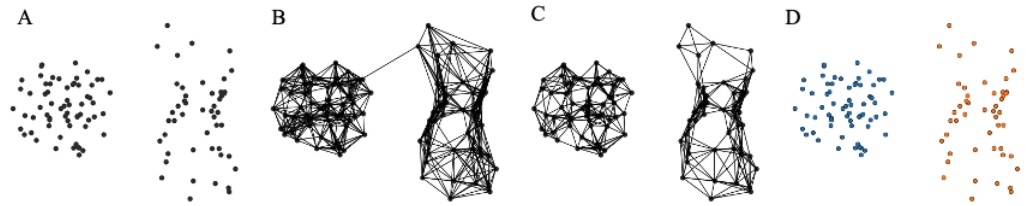

Figure 2: **MBC Algorithm 2 schematic.** (A–B) depicts building a local neighborhood graph for Pattern A in Fig. 1 at the desired sampling scale(s) corresponding to steps 1-4 in Alg. 2, (B–C) depicts to remove unsupported spurious bridges (Alg. 2 steps: 5–6), and (C–D) depicts taking connected components (Alg. 2 steps:7–9); recompute for different $k^*$ from bracket yields the same clusters i.e. a tight bracket of $[2, 2]$ due to the offset-fill distance ratio denoting separability for this dataset.

*where $h_s$ is the (clean) fill distance on $\mathcal{M}_s$, the constants $\underline{C}_s, \overline{C}_s$ depend only on $(d, \underline{c}, \overline{c})$ and the choice of $A, \varepsilon$ (via the uniform clean bounds), and $C_1, C_2 > 0$ are universal. In particular, $H_i := D_k(x_i) = \Theta(h_s) + O(\sigma)$ uniformly on $\mathcal{M}_s$.*

An additional problem is that nearest-neighbor distances based on a single global scale may be too sensitive to density fluctuations. Instead, we compare averages over two scales of neighbors, whose distance distributions, as we show, differ significantly for within- vs. cross-component. To make local decisions robust we employ a two-scale distance-to-measure approach (Chazal et al., 2018) that cancels leading density terms yet reacts to ambient offsets. Fix $\theta > 1$; for a query $z$ and a finite set $T$, let $r_1$ be the $k_1$-th nearest-neighbor distance from $z$ to $T$ with $k_1 \asymp k$, set $k_2 = \#\{u \in T : \|u - z\| \leq \theta r_1\}$, let $a_1$ and $a_2$ be the means of the $k_1$ and $k_2$ smallest distances, and define $\widetilde{d}_\theta(z \to T) = (\theta\, a_1 - a_2)/(\theta - 1)$. as the *two-scale DTM statistic*. When $T$ is drawn from a $d$-dimensional manifold, $\widetilde{d}_\theta$ cancels the first-order $\Theta(h_s)$ bias of the distance-to-measure, leaving a smaller on-manifold remainder, whereas for a point at ambient offset $\Delta_{\text{eff}}$ it grows linearly in $\Delta_{\text{eff}}$. The next proposition makes this separation precise after normalizing by fill distance.

**Proposition 3.6** (Directional two-scale typicality with noise). *Let $H_i = D_k(x_i)$ and form $S_i$ by trimming the kNN list of $x_i$ at radius $c\, H_i$ for fixed $c > 1$ (and, if desired, capping $|S_i|$ by a constant). Fix $\theta > 1$. Then there exist constants $A, B > 0$ depending only on $(d, c, \theta)$ such that, with probability at least $1 - \delta$, the following hold uniformly over $i$:*
- *If $x_j$ lies on the same component as $x_i$, then*

$$\frac{\widetilde{d}_\theta(x_j \to S_i)}{H_i} \leq A\left(\frac{\sigma}{H_i} + \left(\tfrac{k}{n}\right)^{1/d}\right).$$

- *If $x_j$ lies on a different component, let $\Delta_{\text{eff}} := \max\{\Delta - 2\sigma, 0\}$. Then*

$$\frac{\widetilde{d}_\theta(x_j \to S_i)}{H_i} \geq B\frac{\Delta_{\text{eff}}}{H_i} - A\left(\frac{\sigma}{H_i} + \left(\tfrac{k}{n}\right)^{1/d}\right).$$

*Consequently, when $\Delta_{\text{eff}}/h_{\max}$ exceeds a sufficiently large constant (depending on $(d, c, \theta)$ and the local mass bounds), the within- and cross-component distributions of the normalized statistic are separated by a fixed gap.*

## 4 MANIFOLD-BASED CLUSTERING AND THE BRACKET

We now describe a practical clustering pipeline that implements the geometric principles above. Given data $X \in \mathbb{R}^{n \times D}$, we first standardize each feature to zero mean and unit variance. We then estimate an intrinsic dimension $d_{\text{eff}}$ as the smallest number of principal components explaining at least 90% of the variance, capped at 64 components to avoid instability in high dimensions. For a failure budget $\delta \in (0, 1)$, we take a connectivity-safe pilot degree $k^\star = \lceil \log(4n/\delta) \rceil$ and assign a slightly adaptive per-node degree $k_i$ via the pilot radii (ensuring $k_i \geq k^\star$), as detailed in Appendix A.4, to mitigate the effects of non-uniform sampling not accounted for by our theory. We then compute top-$k_i$ Euclidean

neighbors for each point, record local radii $H_i = D_{k_i}(x_i)$, and form the symmetric candidate edge set by keeping $\{i, j\}$ if either $i$ lists $j$ or $j$ lists $i$. Edges are then filtered in two remove-only passes, followed by an add-only step:

(i) The *Euclidean geometric-mean gate* pass enforces scale-adaptive proximity by retaining $\{i, j\}$ only if $\|x_i - x_j\| \leq \sqrt{H_i H_j}$. It discards edges that are too long relative to local sampling density, ensuring connections respect the intrinsic scale.

(ii) The subsequent *triangle support* pass requires a shared nearest neighbor to support an edge between two points, preventing spurious links caused by sampling fluctuations. By Theorem 3.3 and its noisy extension, these two passes eliminate cross-component edges once $\Delta/h_{\max}$ exceeds the corresponding upper threshold.

(iii) Finally, to avoid disconnecting thin structures, e.g., curved manifolds or boundary points, the *add-only rescue* step conservatively reintroduces edges that failed triangle support but are statistically typical of their local neighborhoods. For each node $i$, we form a trimmed local set $S_i \subseteq N_{k_i}(i)$ by discarding neighbors beyond $c\,H_i$ (for a fixed multiplier $c > 1$) and, if necessary, capping $|S_i|$ by a small constant. We then compute a local threshold $\tau_i$ (high local quantile) based on the distribution of neighboring distances in $S_i$: $\tau_i = \mathrm{Quantile}_{q_\tau}\{(\widetilde{d}_\theta(q \to S_i))/H_i : q \in S_i\}$, setting $\theta = 2$ and $q_\tau = 0.90$ in all experiments. An excluded edge $\{i, j\}$ is rescued if and only if neither of its endpoints both look 'typical' with respect to each other's neighborhoods: $\widetilde{d}_\theta(x_j \to S_i)/H_i \leq \tau_i$ and $\widetilde{d}_\theta(x_i \to S_j)/H_j \leq \tau_j$. Theorem 4.2 ensures that, above the noisy-separation threshold, this procedure does not introduce cross-component edges while repairing within-component connectivity near regions of high curvature or at manifold boundaries. Finally, cluster labels are obtained as the connected components of the resulting unweighted graph.

**Uncertainty Bracket.** A key aspect of MBC is that it provides an interpretable measure of uncertainty in the number of clusters. We start from the theoretically motivated degree $k^\star = \lceil A\log(4n/\delta)\rceil$ given by Theorem 3.3, and define $\varepsilon_k = \sqrt{\log(2n/\alpha)/(2k^\star)}$ for confidence level $\alpha \in (0, 1)$. This choice inverts the same binomial Chernoff bounds used in our fill-distance and $k$NN-radius estimates: at the intrinsic sampling radius, the neighbor count around each point has mean $\Theta(k^\star)$ and, with probability at least $1 - \alpha$, deviates by at most $\varepsilon_k k^\star$ uniformly over all $n$ samples. We therefore consider the degree window $k_{\mathrm{low}} = \lceil(1 - \varepsilon_k)k^\star\rceil$ and $k_{\mathrm{high}} = \lceil(1 + \varepsilon_k)k^\star\rceil$. We then recompute only the remove-only base graph (Euclidean gate and triangle support) at these two scales, and set $K_{\mathrm{low}} := \#\mathrm{Comp}(k_{\mathrm{high}})$ and $K_{\mathrm{high}} := \#\mathrm{Comp}(k_{\mathrm{low}})$. Because the candidate edge set is nondecreasing in $k$, the component count is nonincreasing, so $[K_{\mathrm{low}}, K_{\mathrm{high}}]$ forms a monotone bracket capturing all intermediate degrees. Narrow brackets indicate a stable intrinsic scale for the given data; wider brackets signal that the manifold-separation decision is unstable as outlined below.

**Corollary 4.1** (Bracket behavior under the separation threshold). *Assume the setting of Theorem 3.3. For any $k \in [k_{\mathrm{low}}, k_{\mathrm{high}}]$ we can write $k = \lceil A'\log(4n/\delta)\rceil$ with $A' \in [A(1 - \varepsilon_k), A(1 + \varepsilon_k)]$. With thresholds $\overline{C}(A')$ and $\underline{C}(A')$ as in Remark A.6, the $k-$bracket has the property that $\Delta/h_{\max}$ lies in the region*

$$\underline{C}\big(A(1 - \varepsilon_k)\big) \;\leq\; \Delta/h_{\max} \;\leq\; \overline{C}\big(A(1 + \varepsilon_k)\big).$$

Therefore, the learned number of clusters $K_{\mathrm{low}}$ and $K_{\mathrm{high}}$ differ, in effect pushing the uncertainty band on the correct scale for $k^*$ to the threshold in Theorem 3.3. Moreover, since $k_{\mathrm{high}} - k_{\mathrm{low}} \approx 2\varepsilon_k k^\star$ and $k^\star = A\log(4n/\delta)$, the relative width $(k_{\mathrm{high}} - k_{\mathrm{low}})/k^\star$ is bounded at $O(A^{-1/2})$.

Computationally, MBC shares the same leading cost as other $k$NN-based density methods (e.g., DBSCAN, HDBSCAN): constructing the neighborhood graph. In moderate ambient dimension, tree- or graph-based backends yield near-linear scaling in $n$, while in high dimensions brute-force search incurs $O(n^2 D)$ distance evaluations. Once the $k$NN lists are available, all subsequent passes are linear in the number of candidate edges: the Euclidean geometric-mean gate is a single sweep over $O(nk)$ edges, triangle support reduces to intersections of neighbor lists of length $k$, and the add-only rescue is applied only to edges rejected by triangle support, using trimmed neighborhoods $S_i$ of bounded size. The bracket stage requires recomputing only the remove-only graph at two nearby degrees, incurring a constant-factor overhead on top of a single $k$NN construction. Throughout, we fix $\theta = 2$, $q_\tau = 0.90$, the trimming multiplier $c = 4$, and a cap $|S_i| \leq 32$, so that the only exposed scale parameter is $k$, determined by $(n, \delta)$ via the connectivity theory in Theorem 3.3.

---

**Algorithm 1** MBC: Manifold-Based Clustering

---

**Require:** $X \in \mathbb{R}^{n \times D}, \delta, \alpha \in (0, 1)$

1: **Preprocess:** standardize $X$; set $d_{\text{eff}} \leftarrow$ PCA dims; $k^\star \leftarrow \lceil \log(4n/\delta) \rceil$
2: **Pilot / local-$k$:** $H_i^{\text{pilot}} = D_{k^\star}(x_i)$; $H_{\text{ref}} \leftarrow \text{median}\{H_i^{\text{pilot}} > 0\}$; choose $k_i$ around $k^\star$ from $(H_i^{\text{pilot}}, H_{\text{ref}})$; set $H_i = D_{k_i}(x_i)$
3: **$k$NN & candidates:** for each $i$, get $N_i$ (top-$k_i$); $P = \{\{i, j\}: j \in N_i \text{ or } i \in N_j\}$
4: **Euclidean gate:** $E_{\text{eucl}} \leftarrow \{\{i, j\} \in P: \|x_i - x_j\| \leq \sqrt{H_i H_j}\}$
5: **Triangle support:** $E_{\text{tri}} \leftarrow \{\{i, j\} \in E_{\text{eucl}}: |N_i \cap N_j| \geq t_\triangle\}$; $R \leftarrow E_{\text{eucl}} \setminus E_{\text{tri}}$
6: **Add-only DTM rescue:** for each $\{i, j\} \in R$, compute $y_{i \leftarrow j}, y_{j \leftarrow i}$ as normalized two-scale DTM scores w.r.t. $(S_i, H_i)$ and $(S_j, H_j)$ where $S_i = N_i$ capped by $|N_i| = 32$
7:     if $y_{i \leftarrow j} \leq \tau_i$ and $y_{j \leftarrow i} \leq \tau_j$, update $E_{\text{tri}} \leftarrow E_{\text{tri}} \cup \{\{i, j\}\}$
8: **Clusters:** $L \leftarrow \text{ConnectedComponents}(V = [n], E_{\text{tri}})$
9: **$K$-bracket:** choose $\varepsilon_k$ from $(n, \alpha, k^\star)$; recompute graphs at $(1 \pm \varepsilon_k) \cdot k_i$ to get $K_{\min}$ and $K_{\max}$

---

Finally, we justify our algorithm by combining Propositions 3.5–3.6 with the noiseless thresholds. This yields a noisy analog of Theorem 3.3, consistent with the algorithm we implement. Intuitively, when inter-cluster separation is larger than sampling noise, our *remove-only* and *add-only* steps guarantee true cluster identification; when separation is smaller, bridging edges inevitably appear.

**Theorem 4.2** (Noisy separation and safe add-only rescue). *Under the assumptions above (local mass bounds on a fixed small-ball scale, tubular noise of radius $\sigma$, and $k = \lceil A \log(4n/\delta) \rceil$), there exist constants $\underline{C}_\sigma, \overline{C}_\sigma > 0$ such that, with probability at least $1 - \delta$, the Euclidean geometric-mean gate followed by triangle support has no cross-component edges whenever*

$$\frac{\Delta}{h_{\max}} > \overline{C}_\sigma := \overline{C} + C \frac{\sigma}{h_{\max}},$$

*where $\overline{C}$ is the noiseless threshold from Theorem 3.3 and $C > 0$ is universal. Moreover, if one performs an add-only rescue that reinstates an edge $\{i, j\}$ precisely when both directional statistics satisfy $\widetilde{d}_\theta(x_j \to S_i)/H_i \leq \tau_i$ and $\widetilde{d}_\theta(x_i \to S_j)/H_j \leq \tau_j$, with $\tau_i$ the high local quantile of $\{\widetilde{d}_\theta(q \to S_i)/H_i : q \in S_i\}$, then no cross-component edges are added under the same condition. Conversely, if*

$$\frac{\Delta}{h_{\max}} < \underline{C}_\sigma := \underline{C} - C \frac{\sigma}{h_{\max}},$$

*with $\underline{C}$ from Theorem 3.3, then a cross-component edge appears in the $k$NN graph with non-negligible probability.*

**Proof sketch.** By Proposition 3.5, the geometric-mean gate $\sqrt{H_i H_j}$ stays at scale $h_{\max}$ up to $O(\sigma)$, hence if $\Delta/h_{\max} > \overline{C} + C \sigma/h_{\max}$, then every cross pair violates the Euclidean gate and triangle support cannot reintroduce it. For the rescue rule, Proposition 3.6 together with a high local quantile ensures that an off-component point is atypical from at least one side, so mutual acceptance fails. The lower-threshold direction follows from the noisy-overlap argument after replacing $\Delta$ by $\Delta_{\text{eff}}$ as above. See Appendix A.14 for the proofs of the corresponding propositions and theorem.

## 5 EMPIRICAL RESULTS

We evaluate clustering quality across both synthetic and real regimes under a single, scale-aware protocol. *Two Moons* (2D, sampled with noise) and *Concentric Circles* (2D, sampled with noise) probe curvature and nonconvexity; *Gaussian Blobs* (50D, std. 3.0, $K_{\text{true}} = 4$) test high-dimensional separation; *Digits* ($8 \times 8$ grayscale, PCA$\to 50$) and *MNIST* ($28 \times 28$, PCA$\to 50$) stress representation entanglement without learned embeddings. All features are standardized; $d_{\text{eff}}$ is the smallest PCA dimension accounting for $90\%$ variance (cap 64). For MBC, we use the standard configuration outlined in Algorithm 2; see the Appendix 2 for further implementation details. We ran baselines (DBSCAN, OPTICS, BIRCH, HDBSCAN) using library defaults (Pedregosa et al., 2011); details provided in Appendix B.0.1. Metrics are: Adjusted Rand Index (ARI), Normalized Mutual Information (NMI), and mean predicted $K$ over three seeds (Vinh et al., 2010); for MBC we also report

Table 1: **Synthetic, Real And Neural data.** "MBC Bracket" is the median across runs of the monotone component-count interval; other methods do not provide brackets. We **bold** the true number of clusters appearing in our obtained bracket.

| Dataset ($K_{\text{true}}$) | Method | ARI ↑ | NMI ↑ | Mean $K$ | MBC Bracket |
|---|---|---|---|---|---|
| Two Moons (2D; clean, $K_{\text{true}}$=2) | MBC | **1.000** | **1.000** | 2.00 | [**2**, 11] |
| | BIRCH | 0.499 | 0.512 | 3.00 | – |
| | HDBSCAN | 0.487 | 0.548 | 5.67 | – |
| Concentric Circles (2D; clean, $K_{\text{true}}$=2) | MBC | **1.000** | **1.000** | 2.00 | [**2**, 13] |
| | BIRCH | 0.011 | 0.010 | 3.00 | – |
| | HDBSCAN | 0.041 | 0.251 | 10.67 | – |
| Gaussian Blobs (50D; $K_{\text{true}}$=4) | MBC | **1.000** | 0.999 | 4.67 | [**4**, 16] |
| | BIRCH | 0.714 | 0.857 | 3.00 | – |
| | HDBSCAN | **1.000** | **1.000** | 4.00 | – |
| Iris (4 features (tabular); $K_{\text{true}}$=3) | MBC | 0.552 | 0.701 | 4.00 | [1,...**3**,...5] |
| | BIRCH | **0.661** | **0.733** | 3.00 | – |
| | HDBSCAN | 0.139 | 0.347 | 5.00 | – |
| MNIST (PCA→50; $K_{\text{true}}$=10) | MBC | **0.000** | 0.001 | 2.00 | [9, **10**,...14] |
| | BIRCH | **0.000** | 0.001 | 3.00 | – |
| | HDBSCAN | **0.000** | 0.000 | 1.00 | – |
| Fashion–MNIST (PCA→50; $K_{\text{true}}$=10) | MBC | 0.000 | 0.002 | 3.00 | [**10**, 30] |
| | BIRCH | **0.124** | **0.307** | 3.00 | – |
| | HDBSCAN | 0.000 | 0.000 | 1.00 | – |
| V1 (all points; $K_{\text{true}}$=1) | MBC | **1.000** | **1.000** | 1.00 | [**1**, 3] |
| | BIRCH | 0.000 | 0.000 | 222.00 | – |
| | HDBSCAN | 0.000 | 0.000 | 3.00 | – |
| Retina (labeled subset; $K_{\text{true}}$=7) | MBC | -0.001 | 0.005 | 1.00 | [1,...**7**,...14] |
| | BIRCH | 0.671 | 0.782 | 17.00 | – |
| | HDBSCAN | **0.790** | **0.823** | 8.00 | – |
| Retina (all points; $K_{\text{true}}$=7) | MBC | 0.000 | 0.000 | 1.00 | [1,...**7**,...9] |
| | BIRCH | **0.593** | **0.748** | 21.00 | – |
| | HDBSCAN | 0.484 | 0.649 | 28.00 | – |

the median *monotone bracket* $[K_{\text{low}}, K_{\text{high}}]$ computed from two remove-only neighborhood scales (Sec. 4). We report our results for the main comparable benchmarks: HDBSCAN and BIRCH in Table 1. Additional baseline algorithms (DBSCAN, OPTICS, etc.) are reported in Appendix Table 4. We emphasized default parameters to mirror the setting for which MBC was defined—where the ground-truth number of clusters, and thus the correct hyperparameters, are not known a priori.

Our results align with the offset-fill-distance picture: when $\Delta/h$ is large (Moons, Circles), MBC recovers ground truth with narrow brackets; on high-dimensional separated blobs, MBC matches OPTICS and HDBSCAN; when embeddings are entangled (Fashion-MNIST, MNIST) (Deng, 2012; Xiao et al., 2017), all methods degrade yet MBC widens the bracket rather than forcing spurious partitions. This explains the larger brackets on the Two Moons and Concentric Circles datasets, due to noise-induced ambiguity in the sampling. On the synthetic suite, $K_{\text{true}}$ almost always lies within the reported bracket, and the extended noise/anisotropy variants (Appendix Table 4) show the expected widening of the bracket as separation diminishes. As an additional stress test, we construct a heterogeneous-dimensional mixture (helix-plane-sphere) lifted to $D$=10; MBC recovers the three components while density- and centroid-based methods over- or under-split, or mark large fractions as noise (Appendix Fig. 4).

**Neural case study.** To better understand how our algorithm behaves on real world data, where variations in sampling density often obscures distinct clusters, we analyzed neuronal representations from two stages of the visual pathway: **Retina** and primary visual cortex (**V1**). The original study (Dyballa et al., 2024a) argued that retinal responses clustered into functionally coherent groups, roughly 7–8 cell types, whereas responses in cortex (V1) did not. Treating each dataset as a point

cloud with neurons as points, we ran MBC, HDBSCAN, and BIRCH on the labeled Retina, the complete (labeled + unlabeled) Retina, and the V1 data. MBC's cluster estimate was $K=1$ for both datasets (no forced partition), but the brackets diverged: V1 yielded a near-degenerate interval $[1, 3]$, indicating one component at the available sampling scale; Retina produced a substantially wider interval—$[1, 14]$ on the labeled subset and $[1, 9]$ on all points—that consistently contains the true count ($K_{\text{true}}=7$). This suggests a transitional regime: additional sampling (or a slightly finer neighborhood scale) could plausibly cross the separation threshold. In contrast, baselines *forced* clusters—on V1 they returned $K=3$ (HDBSCAN) and $K=222$ (BIRCH); on Retina they returned $K=28$ and $K=21$—without an uncertainty notion. At the *upper* end of the retinal bracket ($K=9$), agreement with labels becomes nontrivial (best ARI $0.205$, best NMI $0.477$), supporting the interpretation that retinal classes are plausibly present but undersampled, whereas V1 remains effectively unclustered, corroborating the physiological understanding of both systems in the original study. Results obtained from our analysis of the neural data are reported in Table 1. Visual summaries appear in Appendix Fig. 6 (separable, transitional, and nonseparable regimes via the $(\Delta, h)$ geometry) and Fig. 5 (comparing baselines to MBC and illustrating bracket-based cluster assignments).

## 6 DISCUSSION

Taken together, the experiments support a simple operational view: recoverability is governed by the offset-to-sampling ratio $\Delta/h$, and what can be said with confidence at the available scale is captured by the monotone bracket. When $\Delta/h$ is large and separability is clear, the bracket is tight and MBC matches the strongest baselines; when embeddings are entangled (Digits, MNIST with linear PCA), all methods struggle, but MBC surfaces this as a widened bracket rather than committing to a spurious partition. The neural case study emphasizes the same point: V1's bracket collapses around one component, whereas Retina's bracket contains the annotated count and admits competitive agreement at its upper end, indicating a transitional, sampling-limited regime. This identification of potentially separable or nonseparable data offers practical guidance when selecting the types of pre-processing pipelines—for instance, choosing the embedding dimensionality or method (linear, such as PCA, or nonlinear, such as UMAP McInnes et al. (2018)) prior to applying a clustering algorithm.

**Limitations.** As with all graph-based clustering, conclusions are representation-dependent: if the embedding entangles classes, increasing neighborhood size cannot manufacture separation. Our empirical coverage—that $K_{\text{true}}$ lies within the bracket on the synthetic suite—relies on the local mass and smoothness conditions used in our analysis. Strong heterogeneity in sampling rate or intrinsic dimension, severe imbalance, or heavy-tailed/non-tubular noise can widen or bias the bracket. Future work will address these challenges by adopting stronger adaptive procedures for local sampling density estimation. Baseline comparisons were kept conservative (primarily relying on library defaults; see Appendix B.0.1); stronger hand-tuning can improve baselines on specific datasets but does not address the core issue that they return a single $K$. When the ground truth clustering is unknown, this opens up the unsupervised learning process to additional bias through arbitrary hyperparameter selection. For example, on the retinal dataset, tuning HDBSCAN over a wide but reasonable range of parameters yields between 5 and 92 clusters, whereas MBC's bracket provides the physiologically motivated $K_{\text{true}} = 7$ clusters (see Appendix B.0.1).

## 7 CONCLUSION

MBC offers a theoretically grounded, parameter-light approach to manifold clustering and recasts the task as a scale-calibrated geometric decision. A local Euclidean gate, a minimal transitivity check, and a quantile two-scale DTM rescue together recover correct components when the separation-to-density ratio $\Delta/h$ is favorable and, otherwise, returns an uncertainty bracket that reflects the sampling limits. The method is robust across curvature and dimension, exposes uncertainty when scale is ambiguous, and degrades transparently as information declines, while remaining simple to implement. In short, MBC makes clustering more accountable to the data: it provides a proposed partition with geometric justification—or an indication that at the given sampling scale the data cannot support one.

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

# A  APPENDIX

## A.1  GEOMETRIC PRELIMINARIES

We briefly review the geometric notions underpinning our analysis: embedded manifolds, reach, tubular neighborhoods, and the tubular-noise model used in the main theorems. Our goal here is to provide the intuition necessary to make sense of the assumptions in Theorem 3.3 and its noisy variants precise for readers who are less familiar. We also recommend the following references: Fefferman et al. (2023); Niyogi et al. (2008).

**Embedded manifolds.** We model high-dimensional data as points sampled from a geometric object $\mathcal{M}$ sitting inside Euclidean space $\mathbb{R}^D$. Formally, a *d-dimensional $C^2$ submanifold* $\mathcal{M} \subset \mathbb{R}^D$ is a subset such that each point $x \in \mathcal{M}$ has a neighborhood that, after a smooth change of coordinates, looks like an open subset of $\mathbb{R}^d$. Intuitively, $\mathcal{M}$ is a smoothly curved $d$-dimensional surface embedded in $\mathbb{R}^D$: for example, a curve ($d{=}1$) in the plane, or a two-dimensional surface ($d{=}2$) in $\mathbb{R}^3$. The ambient Euclidean metric on $\mathbb{R}^D$ induces a natural Riemannian metric on $\mathcal{M}$ (by allowing local patches of the surface to inherit the standard euclidean metric). In the paper we consider a finite union

$$\mathcal{M} = \mathcal{M}_1 \cup \cdots \cup \mathcal{M}_K,$$

where each $\mathcal{M}_k$ is compact, connected, and $C^2$, and different components do not intersect.

**Reach and tubular neighborhoods.** A central notion controlling curvature and global "bottlenecks" is the *reach* of $\mathcal{M}$. The reach $\tau_{\mathcal{M}}$ is the largest radius $r > 0$ such that every point $z$ within Euclidean distance $r$ of $\mathcal{M}$ has a unique nearest point on $\mathcal{M}$. Equivalently, $\tau_{\mathcal{M}}$ is the infimum distance from $\mathcal{M}$ to its *medial axis*, the set of points in $\mathbb{R}^D$ that have two or more nearest neighbors on $\mathcal{M}$. Locally, the reach has the behavoir of being the "inverse" of the curvature. In other words, the principal curvatures of $\mathcal{M}$ are bounded by $1/\tau_{\mathcal{M}}$. For $0 < \sigma < \tau_{\mathcal{M}}$, the *tubular neighborhood* of radius $\sigma$,

$$\mathcal{T}_\sigma(\mathcal{M}) = \{z \in \mathbb{R}^D : \mathrm{dist}(z, \mathcal{M}) \leq \sigma\},$$

is then a well-behaved "thickening" of $\mathcal{M}$ on which the nearest-point projection

$$\pi_{\mathcal{M}} : \mathcal{T}_\sigma(\mathcal{M}) \to \mathcal{M}, \qquad \pi_{\mathcal{M}}(z) = \arg\min_{y \in \mathcal{M}} \|z - y\|$$

is uniquely defined and smooth.

**Tubular noise model.** In practice, data rarely lie exactly on $\mathcal{M}$; measurements are corrupted by ambient noise. Throughout the paper we adopt a *tubular noise* model: a sample $x$ is obtained by first drawing a clean point $y$ from the surface measure on $\mathcal{M}$ and then perturbing it in the ambient space,

$$x = y + \xi, \qquad y \in \mathcal{M}, \quad \|\xi\| \leq \sigma \text{ or } \xi \text{ sub-Gaussian with scale } \sigma,$$

with $\sigma < \tau_{\mathcal{M}}$. The condition $\sigma < \tau_{\mathcal{M}}$ ensures that $x$ remains inside the tubular neighborhood where the projection $\pi_{\mathcal{M}}(x)$ is well-defined and unique. Geometrically, this means that each observed point can be thought of as lying in a small "tube" around $\mathcal{M}$. In the separation results (e.g., Theorem A.14), this model produces an *effective offset* $\Delta_{\mathrm{eff}}$ between components, which differs from the clean offset $\Delta$ by at most $O(\sigma)$, and introduces another additive $O(\sigma)$ slack in the nearest-neighbor radii. Our thresholds and bracket construction are stated in terms of these effective quantities, so that the conclusions remain valid in the presence of such tubular noise.

## A.2 PROOF OF THE THRESHOLD THEOREM FOR $k$NN GRAPHS

### A.2.1 ASSUMPTIONS, LOCAL MASS BOUNDS, AND NOTATION

Let $\mathcal{M} = \mathcal{M}_1 \cup \mathcal{M}_2 \subset \mathbb{R}^D$, where each $\mathcal{M}_i$ is compact, connected, $d$-dimensional, $C^2$, with positive reach. Assume two-sided intrinsic ball-volume growth: for some $0 < c_1 \leq c_2$ and $r_0 > 0$,

$$c_1 r^d \leq \text{Vol}\big(B_{\mathcal{M}_i}(x, r)\big) \leq c_2 r^d, \qquad \forall x \in \mathcal{M}_i, \ 0 < r \leq r_0.$$

Fix $r_* \in (0, r_0]$ and $L \geq 1$ so that, for all $x$ and $r \leq r_*$,

$$B_{\mathcal{M}_i}\Big(x, \frac{r}{L}\Big) \subseteq B(x, r) \cap \mathcal{M}_i \subseteq B_{\mathcal{M}_i}(x, Lr).$$

Let $\mu_i(\cdot) := \text{Vol}((\cdot) \cap \mathcal{M}_i) / \text{Vol}(\mathcal{M}_i)$ be the normalized surface measure. Define local mass constants (valid for all $r \leq r_*$):

$$\underline{c} := \frac{c_1}{L^d}, \qquad \overline{c} := c_2 L^d, \qquad \underline{c} r^d \leq \mu_i\big(B(x, r)\big) \leq \overline{c} r^d.$$

Independently draw $X_1, \ldots, X_{n_1} \overset{\text{i.i.d.}}{\sim} \mu_1$ and $Y_1, \ldots, Y_{n_2} \overset{\text{i.i.d.}}{\sim} \mu_2$; set $n := n_1 + n_2$ and $n_{\min} := \min\{n_1, n_2\}$. For $i = 1, 2$,

$$h_i := \sup_{x \in \mathcal{M}_i} \min_{z \in \{X_1, \ldots, X_{n_i}\}} \|x - z\|,$$

$$h_{\max} := \max\{h_1, h_2\}, \qquad \Delta := \inf\{\|x - y\| : x \in \mathcal{M}_1, \ y \in \mathcal{M}_2\}.$$

We construct the undirected $k$NN graph by symmetrizing the directed $k$-neighbor lists under the ambient Euclidean distance.

### A.2.2 TWO-SIDED FILL-DISTANCE BOUND

**Lemma A.1** (Fill-distance sandwich with explicit dependence on local mass). *For $r_i := (\log(n_i/\delta)/n_i)^{1/d}$ there exist constants*

$$\overline{C_{\text{fill}}} = \overline{C_{\text{fill}}}(d, \underline{c}), \qquad \underline{C_{\text{fill}}} = \underline{C_{\text{fill}}}(d, \overline{c}),$$

*depending only on $(d, \underline{c}, \overline{c})$, such that for all sufficiently large $n_i$ (so that $\overline{C_{\text{fill}}} r_i \leq r_*$ and $\underline{C_{\text{fill}}} r_i \leq r_*$),*

$$\underline{C_{\text{fill}}} r_i \leq h_i \leq \overline{C_{\text{fill}}} r_i \qquad \text{with probability at least } 1 - \frac{\delta}{2}.$$

*One admissible choice is*

$$\overline{C_{\text{fill}}} = \frac{2}{\underline{c}^{1/d}}, \qquad \underline{C_{\text{fill}}} = \frac{1}{2 \overline{c}^{1/d}}.$$

*Proof. Upper bound.* Fix $r \in (0, r_*]$ and cover $\mathcal{M}_i$ by $N(r)$ ambient balls $B(x_j, r)$ with $N(r) \leq C_{\text{cov}} r^{-d}$, where $C_{\text{cov}} = C_{\text{cov}}(d)$. For each center, by $\mu_i(B(x_j, r)) \geq \underline{c} r^d$, the emptiness probability is $\leq \exp(-\underline{c} n_i r^d)$. By the union bound,

$$\Pr\big(\exists j : B(x_j, r) \text{ is empty}\big) \leq C_{\text{cov}} r^{-d} \exp(-\underline{c} n_i r^d).$$

Choose $r$ so that $\underline{c} n_i r^d = 2 \log(n_i/\delta)$, i.e.

$$r = \frac{2^{1/d}}{\underline{c}^{1/d}} \Big(\frac{\log(n_i/\delta)}{n_i}\Big)^{1/d}.$$

Then

$$\Pr\big(\exists \text{ empty } B(x_j, r)\big) \leq \frac{C_{\text{cov}} \underline{c}}{2} \cdot \frac{\delta^2}{n_i \log(n_i/\delta)} \leq \frac{\delta}{4} \quad \text{for all large } n_i.$$

If no cover ball is empty, each $B(x_j, r)$ contains a sample; any $x \in \mathcal{M}_i$ lies within $r$ of some $x_j$, hence within $2r$ of a sample; therefore $h_i \leq 2r$. With the chosen $r$, this gives

$$h_i \leq \overline{C_{\text{fill}}} r_i, \qquad \overline{C_{\text{fill}}} := \frac{2}{\underline{c}^{1/d}}.$$

*Lower bound.* Let $\mathcal{P}$ be a packing by $M(r)$ disjoint ambient balls of radius $r/2$ centered on $\mathcal{M}_i$, with $M(r) \geq C_{\mathrm{pack}} \, r^{-d}$ and $C_{\mathrm{pack}} = C_{\mathrm{pack}}(d)$. If every such ball contains a sample, then $h_i < r$; conversely, if at least one is empty then $h_i \geq r/2$. For any packed ball $B$, $\mu_i(B) \leq \overline{c}(r/2)^d$, so

$$\Pr(B \text{ is occupied}) \ \leq \ n_i \, \overline{c} \left(\frac{r}{2}\right)^d.$$

By the union bound over $M(r)$ disjoint balls,

$$\Pr\left(\text{all packed balls occupied}\right) \ \leq \ M(r) \, n_i \, \overline{c} \left(\frac{r}{2}\right)^d \ \leq \ \frac{C_{\mathrm{pack}} \, \overline{c}}{2^d} \, n_i.$$

Choose $r = \underline{C}_{\mathrm{fill}} \, r_i$ with $\underline{C}_{\mathrm{fill}} := \frac{1}{2 \, \overline{c}^{1/d}}$. Then $n_i \, \overline{c} \, (r/2)^d = \frac{1}{2} \log(n_i/\delta)$ and

$$\Pr\left(\text{all packed balls occupied}\right) \ \leq \ \frac{C_{\mathrm{pack}}}{2^{d+1}} \cdot \frac{n_i}{\log(n_i/\delta)} \ \leq \ \frac{\delta}{4} \quad \text{for all large } n_i.$$

With probability at least $1 - \delta/4$ some packed ball is empty, whence $h_i \geq r/2$; our definition of $\underline{C}_{\mathrm{fill}}$ includes this factor, so $h_i \geq \underline{C}_{\mathrm{fill}} \, r_i$. Combining the two tails (upper and lower) across $i = 1, 2$ yields the claim with probability $\geq 1 - \delta/2$. $\qquad\square$

### A.2.3 Uniform concentration of $k$NN radii at the samples

**Lemma A.2** (Uniform $k$NN upper bound). *Fix $\varepsilon \in (0, 1)$ and choose*

$$k \ = \ \left\lceil A \log\left(\frac{4n}{\delta}\right) \right\rceil, \qquad A \ \geq \ \frac{3}{\varepsilon^2}.$$

*Let $D_k(Z)$ be the distance from a sample $Z$ to its $k$th nearest neighbor among all $n - 1$ points. Then, with probability at least $1 - \frac{\delta}{2}$, simultaneously for all samples $Z$ from component $\mathcal{M}_i$,*

$$D_k(Z) \ \leq \ \frac{1}{(1 - \varepsilon)^{1/d}} \left(\frac{2k}{n_{\min} \, \underline{c}}\right)^{1/d}.$$

*Proof.* Fix a sample $Z \in \mathcal{M}_i$. For any $r \leq r_*$, the count

$$S(r) \ := \ \#\{j \neq Z : \|Z_j - Z\| \leq r\}$$

is $\mathrm{Bin}(n - 1, p(r))$ with $p(r) \geq \underline{c} r^d$ (we only need same-component mass to lower bound $p(r)$). Let $r$ satisfy $(n_i - 1)\underline{c}r^d = k$. Then $\mathbb{E}[S(r)] \geq k$, and Chernoff's lower tail gives

$$\Pr\left(S(r) \leq (1 - \varepsilon) \, k\right) \ \leq \ \exp\left(-\frac{\varepsilon^2}{2} \, k\right) \ \leq \ \frac{\delta}{4n},$$

by the choice of $k$. Thus $S(r) \geq (1 - \varepsilon)k$ with probability $\geq 1 - \delta/(4n)$; equivalently,

$$D_k(Z) \ \leq \ \frac{r}{(1 - \varepsilon)^{1/d}} \ = \ \frac{1}{(1 - \varepsilon)^{1/d}} \left(\frac{k}{(n_i - 1) \, \underline{c}}\right)^{1/d}.$$

Apply a union bound over all $n$ samples and use $n_i - 1 \geq n_{\min}/2$ to conclude

$$D_k(Z) \ \leq \ \frac{1}{(1 - \varepsilon)^{1/d}} \left(\frac{2k}{n_{\min} \, \underline{c}}\right)^{1/d} \quad \text{for all samples } Z \text{ with probability at least } 1 - \frac{\delta}{2}.$$

$\qquad\square$

**From $D_k$ to a multiple of $h_{\max}$.** By Lemma A.1, for the worse component,

$$h_{\max} \ \geq \ \underline{C}_{\mathrm{fill}} \left(\frac{\log(n_{\min}/\delta)}{n_{\min}}\right)^{1/d}.$$

Combining with Lemma A.2 and $k = A \log(4n/\delta)$ yields, uniformly over all samples $Z$,

$$\frac{D_k(Z)}{h_{\max}} \ \leq \ \frac{1}{(1 - \varepsilon)^{1/d} \, \underline{C}_{\mathrm{fill}}} \left(\frac{2A \, \log(4n/\delta)}{\underline{c} \, \log(n_{\min}/\delta)}\right)^{1/d} \ = \ \frac{1}{(1 - \varepsilon)^{1/d} \, \underline{C}_{\mathrm{fill}}} \left(\frac{2A \, R}{\underline{c}}\right)^{1/d}.$$

**Proposition A.3** (No-bridge regime). *Define*

$$\overline{C} \; := \; \frac{1}{(1 - \varepsilon)^{1/d} \, \underline{C}_{\text{fill}}} \left( \frac{2 A \, R}{\underline{c}} \right)^{1/d}.$$

*If $\Delta > \overline{C} \, h_{\max}$, then the (symmetrized) kNN graph contains no edge connecting $\mathcal{M}_1$ and $\mathcal{M}_2$.*

*Proof.* For any sample $Z$ and any point $W$ on the other manifold, $\|Z - W\| \geq \Delta > \overline{C} \, h_{\max} \geq D_k(Z)$, so $W$ cannot be among the $k$ nearest neighbors of $Z$. $\qquad \square$

### A.2.4   BRIDGING AT SMALL SEPARATION

**Proposition A.4** (Bridge existence under controlled crowding). *Fix $a \in (0, 1/8)$ and write $B(a) := 1 + 2a$. Assume $B(a) \, \Delta \leq r_*$. Define*

$$\overline{C_{\text{fill}}} \;\; \text{as in Lemma A.1,} \qquad \underline{C} \; := \; \frac{1}{\overline{C_{\text{fill}}}} \left( \frac{A \, R}{4 \, \overline{c} \, B(a)^d} \right)^{1/d}.$$

*If $\Delta < \underline{C} \, h_{\max}$, then with probability at least*

$$1 \; - \; 2 \exp\!\left( - \underline{c} \, a^d \, n_{\min} \Delta^d \right) \; - \; \exp(-\gamma k)$$

*(for some absolute $\gamma > 0$) the kNN graph contains a cross-component edge.*

*Proof.* Let $(x_0, y_0) \in \mathcal{M}_1 \times \mathcal{M}_2$ realize $\|x_0 - y_0\| = \Delta$ and consider the intrinsic caps

$$U := B_{\mathcal{M}_1}(x_0, a\Delta), \qquad V := B_{\mathcal{M}_2}(y_0, a\Delta).$$

By the lower mass bound, $\mu_1(U), \mu_2(V) \geq \underline{c} \, (a\Delta)^d$, so

$$\Pr(U \text{ empty}) \; \leq \; e^{-\underline{c} \, a^d \, n_1 \, \Delta^d}, \qquad \Pr(V \text{ empty}) \; \leq \; e^{-\underline{c} \, a^d \, n_2 \, \Delta^d}.$$

Hence with probability at least $1 - 2e^{-\underline{c} \, a^d \, n_{\min} \Delta^d}$ there exist samples $x \in U$ and $y \in V$, and

$$\|x - y\| \; \leq \; \|x - x_0\| + \|x_0 - y_0\| + \|y_0 - y\| \; \leq \; B(a) \, \Delta.$$

Let

$$S_x \; := \; \#\{X_j \in \mathcal{M}_1 : \; \|X_j - x\| \leq B(a) \, \Delta\}.$$

By the upper mass bound,

$$\mathbb{E}[S_x] \; \leq \; (n_1 - 1) \, \overline{c} \, \big( B(a) \, \Delta \big)^d.$$

Assume $h_{\max} = h_1$ (the harder case). If we write $\Delta = \underline{C} \, h_{\max}$ and use the *upper* fill bound from Lemma A.1,

$$h_{\max} \; \leq \; \overline{C_{\text{fill}}} \left( \frac{\log(n_{\min}/\delta)}{n_{\min}} \right)^{1/d},$$

then

$$\mathbb{E}[S_x] \; \leq \; \overline{c} \, B(a)^d \, \big( \underline{C} \, \overline{C_{\text{fill}}} \big)^d \, \log\!\left( \frac{n_{\min}}{\delta} \right).$$

With $k = A \log(4n/\delta) = A \, R \, \log(n_{\min}/\delta)$, the condition

$$\overline{c} \, B(a)^d \, \big( \underline{C} \, \overline{C_{\text{fill}}} \big)^d \; \leq \; \frac{A \, R}{4}$$

ensures $\mathbb{E}[S_x] \leq k/4$ and, by Chernoff, $\Pr(S_x \geq k/2) \leq e^{-\gamma k}$ for some absolute $\gamma > 0$. On this event, fewer than $k$ same-component points lie inside $B(x, B(a)\Delta)$ while $y$ also lies in this ball, so at least one of the $k$ nearest neighbors of $x$ is cross-component. Solving the displayed condition for $\underline{C}$ yields the stated value. $\qquad \square$

### A.2.5 THRESHOLD THEOREM

**Theorem A.5** (Critical separation for the symmetrized $k$NN graph)**.** *Fix $\varepsilon \in (0, 1)$, $a \in (0, 1/8)$, and choose $k = \lceil A \log(4n/\delta) \rceil$ with $A \geq 3/\varepsilon^2$. Let $\overline{C}$ be as in Proposition A.3 and $\underline{C}$ as in Proposition A.4. Then, with probability at least $1 - \delta$ (up to the explicit tails in Proposition A.4):*

1. ***(Disconnected regime)** If $\dfrac{\Delta}{h_{\max}} > \overline{C}$, the $k$NN graph contains no cross-component edge.*

2. ***(Bridged regime)** If $B(a) \Delta \leq r_*$ and $\dfrac{\Delta}{h_{\max}} < \underline{C}$, the $k$NN graph contains at least one cross-component edge with probability at least*

$$1 - 2 \exp\left(-\underline{c}\, a^d\, n_{\min}\, \Delta^d\right) - \exp(-\gamma k).$$

*Remark* A.6. On the constants $\overline{C}$ and $\underline{C}$ With the definitions and choices in Section A.2 (in particular, $k = \lceil A \log(4n/\delta) \rceil$, $R = \log(4n/\delta)/\log(n_{\min}/\delta)$, $B = 1 + 2a$, and the local mass bounds $\underline{c}, \overline{c}$), the threshold constants that govern the disconnected and bridged regimes are

$$\overline{C} = \frac{2}{(1-\varepsilon)^{1/d}} \left(\frac{2\, A\, R\, \overline{c}}{\underline{c}}\right)^{1/d}, \qquad \underline{C} = \left(\frac{A\, R\, \underline{c}}{2^{\,d+2}\, \overline{c}\, B^d}\right)^{1/d}.$$

**Monotonicity and interpretation.** Both $\overline{C}$ and $\underline{C}$ scale like $A^{1/d}$: increasing $k$ (via $A$) makes the no-bridge *condition* stricter (larger $\overline{C}$) and the bridge *condition* easier to meet (larger $\underline{C}$), consistent with the fact that larger $k$ adds edges. The ratio $\overline{c}/\underline{c}$ measures geometry/density skew: $\overline{C}$ grows with $(\overline{c}/\underline{c})^{1/d}$, while $\underline{C}$ shrinks with $(\overline{c}/\underline{c})^{1/d}$, reflecting that heavier local mass and distortion increase same-component crowding. The guard buffer $B$ appears only in $\underline{C}$ (as $1/B$ after the $d$-th root), encoding that a larger buffer makes it harder to force a cross edge. The dependence on $d$ is via $1/d$-powers, so in higher dimensions both constants vary more gently with $A$, $B$, and $\overline{c}/\underline{c}$.

**Practical choices for constants.** For balanced sampling one has $R \approx 1$. Choosing a moderate tail slack $\varepsilon = \frac{1}{2}$ gives the benign factor $(1 - \varepsilon)^{-1/d} = 2^{1/d}$. In typical practice $k = \Theta(\log(n/\delta))$ with a small constant, so $A$ can be taken in a tight range, and one uses a small collar $a$ so $B = 1 + 2a \approx 1$ while still meeting the small-radius condition. Under these settings, and in benign geometry where $\overline{c}/\underline{c} \approx 1$, the formulas simplify to the order-one approximations

$$\overline{C} \approx 2^{1/d} \left(4A\right)^{1/d}, \qquad \underline{C} \approx \frac{1}{2B} \left(\frac{A\, \underline{c}}{\overline{c}}\right)^{1/d},$$

so taking $A \approx 1$, $B \approx 1$, and $\overline{c}/\underline{c} \approx 1$ leaves both thresholds at a natural, dimension-controlled constant scale, with their gap dominated by the simple $1/(2B)$ factor in $\underline{C}$.

### A.2.6 COROLLARIES FOR KERNEL GRAPHS

**Corollary A.7** (Gaussian (RBF) kernel: inter-manifold suppression and activation)**.** *Fix a bandwidth $\sigma > 0$ and define*

$$w(x, y) := \exp\left(-\frac{\|x - y\|^2}{\sigma^2}\right), \qquad W_{12} := \sum_{x \in S_1} \sum_{y \in S_2} w(x, y),$$

*where $S_1, S_2$ are the sample sets on $\mathcal{M}_1, \mathcal{M}_2$. On the high-probability event of Theorem A.5 the following hold.*

1. ***(Disconnected regime)** If $\Delta > \overline{C}\, h_{\max}$, then for every $x \in S_1$ and $y \in S_2$,*

$$\|x - y\| \geq \Delta \qquad \Longrightarrow \qquad w(x, y) \leq \exp\left(-\frac{\Delta^2}{\sigma^2}\right),$$

*and hence*

$$W_{12} \leq n_1 n_2 \exp\left(-\frac{\Delta^2}{\sigma^2}\right).$$

2. **(Bridged regime)** *Assume the small-radius condition $B \Delta \leq r_*$ and suppose $\Delta < \underline{C} \, h_{\max}$. Then, with probability at least*

$$1 \; - \; 2 \exp\!\big(\underline{c} \, a^d \, n_{\min} \, \Delta^d\big) \; - \; \exp(-\gamma k),$$

*there exist $x \in S_1$ and $y \in S_2$ such that*

$$\|x - y\| \; \leq \; B \, \Delta \qquad \Longrightarrow \qquad w(x, y) \; \geq \; \exp\!\Big(\!-\frac{B^2 \, \Delta^2}{\sigma^2}\Big),$$

*and consequently*

$$W_{12} \; \geq \; \exp\!\Big(\!-\frac{B^2 \, \Delta^2}{\sigma^2}\Big).$$

*Proof.* On the event of Theorem A.5, the no-bridge regime ensures all cross-component pairs are at distance at least $\Delta$; the displayed upper bound follows by monotonicity of $r \mapsto \exp(-r^2/\sigma^2)$, and the bound on $W_{12}$ follows by summing over $n_1 n_2$ pairs.

In the bridged regime, Proposition A.4 guarantees the existence of a cross pair with $\|x - y\| \leq B \Delta$ with the stated probability. The lower bound follows by monotonicity and by retaining one such pair in the sum defining $W_{12}$. $\qquad\square$

### A.3  DTM AND NOISY THRESHOLD CRITERION

#### A.3.1  TUBULAR NOISE MODEL AND A TWO-SCALE AVERAGED-DISTANCE STATISTIC

We adopt the tubular-noise model from the main text. For each component $\mathcal{M}_s \subset \mathbb{R}^D$ (compact, connected, $C^2$, reach $\tau_s > 0$), each observed sample $x$ is generated as

$$x \; = \; \pi_{\mathcal{M}_s}(x) \; + \; \xi, \tag{1}$$

where $\pi_{\mathcal{M}_s}$ is the nearest-point projection (well-defined whenever $\|\xi\| < \tau_s$) and $\xi$ is either (i) almost surely bounded with $\|\xi\| \leq \sigma < \tau_{\min} := \min_s \tau_s$, or (ii) sub-Gaussian with scale $\sigma$ truncated to $\|\xi\| < \tau_{\min}$.

**Noise-sparsity regime.** We work under

$$\sigma \; \leq \; c_{\mathrm{noise}} \, h_{\max} \tag{2}$$

for a fixed constant $c_{\mathrm{noise}} \in (0, 1)$, so that $k$NN radii are at least of order $\sigma$ and the local small-ball law remains $d$-dimensional up to absolute constants. All constants below may depend on $c_{\mathrm{noise}}$.

**Definition A.8** (Two-scale averaged-distance statistic). Let $T$ be a finite subset of $\mathbb{R}^D$ and $z \in \mathbb{R}^D$. For $m \in \{1, \ldots, |T|\}$ let $r_m(z \mid T)$ be the $m$th nearest-neighbor distance from $z$ to $T$, and define

$$\bar{d}_m(z \mid T) \; := \; \frac{1}{m} \sum_{\ell=1}^{m} r_\ell(z \mid T).$$

Fix a scale factor $\theta > 1$. Given an integer $k_1 \geq 1$, set

$$k_2 \; := \; \#\big\{u \in T : \|u - z\| \leq \theta \, r_{k_1}(z \mid T)\big\}, \qquad \widetilde{d}_\theta(z \to T) \; := \; \frac{\theta \, \bar{d}_{k_1}(z \mid T) \; - \; \bar{d}_{k_2}(z \mid T)}{\theta - 1}.$$

Given the global $k$ from the $k$-choice in Section A.2 (namely $k = \lceil A \log(4n/\delta) \rceil$ with $A \geq 3/\varepsilon^2$), let $H_i := D_k(x_i)$ and define the trimmed neighbor set

$$S_i \; := \; \big\{q \in N_k(i) : \|x_q - x_i\| \leq c_{\mathrm{trim}} H_i\big\}, \qquad |S_i| \leq S_{\max}, \tag{3}$$

for fixed constants $c_{\mathrm{trim}} > 1$ and $S_{\max} \in \mathbb{N}$. Trimming ensures bounded differences for the per-node statistics used below.

### A.3.2 Tubular small-ball probabilities and noisy $k$NN radii

Throughout, let the local mass bounds from Section A.2 hold on radii $\leq r_*$:

$$\underline{c}\, r^d \;\leq\; \mu_s\big(B(x,r)\big) \;\leq\; \overline{c}\, r^d, \qquad \text{for all } x \in \mathcal{M}_s,\ 0 < r \leq r_*,$$

where $\mu_s$ is the normalized surface measure on $\mathcal{M}_s$.

**Lemma A.9** (Tubular local-mass sandwich). *Fix $\mathcal{M}_s$ and a point $x = \pi_{\mathcal{M}_s}(x)+\xi$ with $\|\xi\| \leq \sigma < \tau_s$. There exist radii $0 < r_{\mathrm{low}} \leq r_\bullet \leq r_*$ with*

$$r_{\mathrm{low}} \;:=\; 2\sigma, \qquad r_\bullet \;:=\; \min\{\, r_* - \sigma,\ \tau_s/2 \,\}, \tag{4}$$

*and constants*

$$\underline{c}_\sigma \;:=\; \underline{c}\,\big(1 - C\sigma/\tau_s\big), \qquad \overline{c}_\sigma \;:=\; \overline{c}\,\big(1 + C\sigma/\tau_s\big),$$

*such that, for all $r \in [r_{\mathrm{low}},\, r_\bullet]$,*

$$\underline{c}_\sigma\, r^d \;\leq\; \Pr\big(\|X - x\| \leq r\big) \;\leq\; \overline{c}_\sigma\, r^d, \tag{5}$$

*where $X$ is an independent sample from the tubular model on $\mathcal{M}_s$ and $C > 0$ is an absolute constant.*

*Proof.* Write $m := \pi_{\mathcal{M}_s}(x)$ and work in normal coordinates at $m$. Any sample $X$ can be written as $X = M + \zeta$ with $M \sim \mu_s$ on $\mathcal{M}_s$ and $\zeta$ an independent noise with $\|\zeta\| < \tau_s$. For any $r \geq 2\sigma$ and any $\|\zeta\| \leq \sigma$,

$$B_{\mathcal{M}_s}(m,\, r - \|\zeta\|) \;\subseteq\; \{u \in \mathcal{M}_s :\, \|u + \zeta - x\| \leq r\} \;\subseteq\; B_{\mathcal{M}_s}(m,\, r + \|\zeta\|).$$

Integrating the indicator $\mathbf{1}\{\|M + \zeta - x\| \leq r\}$ over $\zeta$ and using that $r \pm \|\zeta\| \in [r/2,\, 3r/2]$ when $r \geq 2\sigma$ shows that $\Pr(\|X - x\| \leq r)$ is equivalent, up to multiplicative constants independent of $x$ and $r$, to $\mu_s(B_{\mathcal{M}_s}(m, r))$ at scales $\leq r_*$. The Jacobian bounds for the exponential map on radii $\leq r_*$ and the truncation $\|\zeta\| \leq \sigma$ produce only a relative $(1 \pm C\sigma/\tau_s)$ distortion. Absorbing fixed factors into $\underline{c}_\sigma, \overline{c}_\sigma$ yields equation 5. $\qquad\square$

**Lemma A.10** (Noisy $k$NN radius concentration (uniform at the samples)). *Let $x$ lie on component $\mathcal{M}_s$ under the tubular model with $\sigma < \tau_s$ and assume equation 2. Let $k = \lceil A\log(4n/\delta)\rceil$ with $A \geq 3/\varepsilon^2$. There exist $C_1, C_2 > 0$ such that, with probability at least $1 - \delta$,*

$$\left(\frac{k}{(n_s - 1)\,\overline{c}_\sigma}\right)^{1/d} - \; C_1\,\sigma \;\leq\; D_k(x) \;\leq\; \left(\frac{k}{(n_s - 1)\,\underline{c}_\sigma}\right)^{1/d} + \; C_2\,\sigma, \tag{6}$$

*uniformly over all samples $x$ drawn from $\mathcal{M}_s$. In particular $D_k(x) = \Theta((k/n_s)^{1/d})$ and, for $k \asymp \log n$, $D_k(x) \asymp h_s$.*

*Proof.* Let $r_0(x)$ solve $(n_s - 1)\Pr(\|X - x\| \leq r_0) = k$. By Lemma A.9, provided $r_0 \in [2\sigma, r_\bullet]$,

$$\left(\tfrac{k}{(n_s-1)\,\overline{c}_\sigma}\right)^{1/d} \;\leq\; r_0(x) \;\leq\; \left(\tfrac{k}{(n_s-1)\,\underline{c}_\sigma}\right)^{1/d}.$$

In the regime equation 2 and $k \gtrsim \log n$, one has $r_0 \gtrsim (k/n_s)^{1/d} \gtrsim h_s \gtrsim \sigma$, hence $r_0 \in [2\sigma, r_\bullet]$ for all large $n_s$. For fixed $x$, $S(r) := \#\{j \neq x :\, \|X_j - x\| \leq r\}$ is $\mathrm{Bin}(n_s - 1, p(r))$ with $p(r) = \Pr(\|X - x\| \leq r)$. At $r = r_0(x)$, $\mathbb{E}S(r_0) = k$. Chernoff implies

$$\Pr\big(|S(r_0) - k| \geq \varepsilon k\big) \;\leq\; 2\exp(-c\,\varepsilon^2 k).$$

On the complement, $(1 - \varepsilon)r_0 \leq D_k(x) \leq (1 + \varepsilon)r_0$. A union bound over all $x$ together with $k = \lceil A\log(4n/\delta)\rceil$ (and $A \geq 3/\varepsilon^2$) yields the claim; the additive $O(\sigma)$ terms follow from the $(1 \pm C\sigma/\tau_s)$ perturbation of $\underline{c}, \overline{c}$ in Lemma A.9. $\qquad\square$

### A.3.3 TWO-SCALE STATISTIC: BIAS CANCELLATION AND OFFSET RESPONSE

**Lemma A.11** (Bias cancellation on-manifold). *Let $T$ be i.i.d. samples from $\mathcal{M}_s$ satisfying the local mass bounds on radii $\leq r_*$. Fix $\theta > 1$ and take $k_1 \asymp k$ with $k = \lceil A \log(4n/\delta) \rceil$. There exist constants $A_0, B_0 > 0$ (depending on $d, \underline{c}, \overline{c}, \theta$) such that, with probability at least $1 - \delta$, uniformly for $z$ on $\mathcal{M}_s$,*

$$\left| \widetilde{d}_\theta(z \to T) - \beta_s(z) \right| \; \leq \; A_0 \left( \frac{k}{n_s} \right)^{1/d} h_s, \qquad \beta_s(z) = O\big( h_s^{1+2/d} \big). \tag{7}$$

*Proof.* Write $F(r) := \Pr(\|X - z\| \leq r)$ for $X \sim \mu_s$. In normal coordinates (valid for $r \leq r_*$),
$$F(r) \; = \; \lambda_d r^d \big( 1 + \kappa_2 r^2 + O(r^3) \big),$$
with $\lambda_d \in [\underline{c}, \overline{c}]$ and $\kappa_2$ depending on curvature. The quantile $Q(u) := F^{-1}(u)$ satisfies $Q(u) = (u/\lambda_d)^{1/d} \big( 1 + \tilde{\kappa}_2 u^{2/d} + O(u^{3/d}) \big)$ for small $u$. For $m = o(n_s)$,

$$\mathbb{E} \, \bar{d}_m(z \mid T) = \frac{n_s}{m} \int_0^{m/n_s} Q(u) \, du = c_d \left( \frac{m}{n_s} \right)^{1/d} + b \left( \frac{m}{n_s} \right)^{(1+2/d)} + O\big( (m/n_s)^{1+3/d} \big),$$

with $c_d > 0$ and $b$ depending on curvature. Put $\alpha := (k_1/n_s)^{1/d}$. One has $k_2/n_s = \theta^d k_1/n_s \, (1 + O(\alpha^2))$, and $r_{k_2} = \theta r_{k_1} (1 + O(\alpha^2))$. Therefore

$$\mathbb{E} \, \widetilde{d}_\theta(z \to T) = \frac{\theta \, \mathbb{E} \, \bar{d}_{k_1} - \mathbb{E} \, \bar{d}_{k_2}}{\theta - 1} = \frac{b \, \alpha^{1+2/d} \big[ \theta - \theta^{1+2/d} \big]}{\theta - 1} + O(\alpha^{1+3/d}),$$

so the linear term in $\alpha$ cancels. Since $\alpha \asymp (k/n_s)^{1/d} \asymp h_s$ and $\alpha^{1+2/d} = \Theta((k/n_s)^{1/d} h_s)$, the bias is $O(h_s^{1+2/d})$. Concentration of $\bar{d}_m$ is $O(\alpha \sqrt{\log(n/\delta)/k})$, dominated by $\alpha^{1+2/d}$ for $k \asymp \log n$. A covering at scale $r_{k_1}$ and a union bound give the uniform bound with probability $\geq 1 - \delta$. $\qquad\square$

**Lemma A.12** (Offset response). *Let $z$ satisfy $\mathrm{dist}(z, \mathcal{M}_s) = \Delta_{\mathrm{eff}}$. For any trimmed $S \subseteq N_k(i)$ with equation 3 and any $\theta > 1$, there exists $B_0' > 0$ (depending on $d, \underline{c}, \overline{c}, \theta, c_{\mathrm{trim}}$) such that, with probability at least $1 - \delta$,*

$$\widetilde{d}_\theta(z \to S) \; \geq \; B_0' \, \Delta_{\mathrm{eff}} - C_\sigma \, \sigma - A_0 \left( \frac{k}{n_s} \right)^{1/d} h_s. \tag{8}$$

*Proof.* For any $u \in S$,
$$\|z - u\| \; \geq \; \mathrm{dist}(z, \mathcal{M}_s) - \mathrm{dist}(u, \mathcal{M}_s) \; \geq \; \Delta_{\mathrm{eff}} - \|\xi(u)\| \; \geq \; \Delta_{\mathrm{eff}} - \sigma.$$
Thus $\bar{d}_{k_1}(z \mid S) \geq \Delta_{\mathrm{eff}} - \sigma$ and $\bar{d}_{k_2}(z \mid S) \geq \Delta_{\mathrm{eff}} - \sigma$, hence

$$\widetilde{d}_\theta(z \to S) \; = \; \frac{\theta \, \bar{d}_{k_1} - \bar{d}_{k_2}}{\theta - 1} \; \geq \; \Delta_{\mathrm{eff}} - \sigma.$$

Curvature and trimming affect this by a fixed factor $B_0' \in (0, 1]$; sampling fluctuations contribute the $A_0((k/n_s)^{1/d} h_s)$ term via Lemma A.11, giving equation 8. $\qquad\square$

**Lemma A.13** (Quantile stability). *Fix $i$ and let $Z_q := \widetilde{d}_\theta(x_q \to S_i)/H_i$ for $q \in S_i$. Let $\tau_i$ be the empirical $q_\tau$-quantile with $q_\tau \in (0.9, 1)$. There exists $C_\tau > 0$ such that, with probability at least $1 - \delta$,*

$$\big| \, \tau_i - Q_i(q_\tau) \, \big| \; \leq \; C_\tau \sqrt{\frac{\log(n/\delta)}{|S_i|}}, \tag{9}$$

*where $Q_i$ is the population quantile of $Z_q$ when $q$ ranges over same-component neighbors in $S_i$.*

*Proof.* Condition on $S_i$. Each $Z_q \in [0, c_{\mathrm{trim}}]$ since $S_i$ is trimmed. Replacing one neighbor $q \in S_i$ changes the multiset $\{Z_q\}$ in at most one coordinate within a bounded interval, so the empirical CDF varies by at most $1/|S_i|$. McDiarmid's inequality yields

$$\Pr\Big( |\tau_i - \mathbb{E}[\tau_i \mid S_i]| \geq t \ \Big| \ S_i \Big) \; \leq \; 2 \exp\Big( -\frac{2t^2 \, |S_i|}{L^2} \Big),$$

for some $L \lesssim c_{\mathrm{trim}}$. Set $t = C_\tau \sqrt{\log(n/\delta)/|S_i|}$ and absorb the bias $|\mathbb{E}[\tau_i \mid S_i] - Q_i(q_\tau)|$ into $C_\tau$ using standard quantile smoothness under the two-sided mass bound. A union bound over $i$ gives equation 9. $\qquad\square$

A.3.4 NOISY SEPARATION THRESHOLDS AND SAFETY OF ADD-ONLY RESCUE

Let $\overline{C}$ and $\underline{C}$ be the noiseless threshold constants defined in Section A.2:

$$\overline{C} = \frac{1}{(1-\varepsilon)^{1/d} \, \underline{C}_{\text{fill}}} \left(\frac{2A\,R}{\underline{c}}\right)^{1/d}, \qquad \underline{C} = \frac{1}{\overline{C}_{\text{fill}}} \left(\frac{A\,R}{4\,\overline{c}\,B(a)^d}\right)^{1/d},$$

with $R = \log(4n/\delta)/\log(n_{\min}/\delta) \approx 1$, $\underline{C}_{\text{fill}} = 1/(2\,\overline{c}^{1/d})$, $\overline{C}_{\text{fill}} = 2/\underline{c}^{1/d}$, and $B(a) = 1 + 2a$.

**Theorem A.14** (Noisy thresholds and safety of add-only rescue). *Under the tubular-noise model equation 1–equation 2 and with $k = \lceil A \log(4n/\delta) \rceil$ (the factor $4n$ inside $\log(4n/\delta)$ originates from a union bound over $n$ sample points and two Chernoff tails per point), there exist constants $\kappa_+, \kappa_- > 0$ (depending only on $d, \underline{c}, \overline{c}, \theta, c_{\text{trim}}$) such that, with probability at least $1 - \delta$:*

1. ***(Upper/no-bridge)** If*

$$\frac{\Delta}{h_{\max}} > \overline{C} + \kappa_+ \frac{\sigma}{h_{\max}}, \tag{10}$$

   *then the Euclidean geometric-mean gate followed by triangle support yields* no *cross-component edges.*

2. ***(Add-only rescue is safe)** Under equation 10, the add-only rescue that reinstates $\{i, j\}$ iff $\widetilde{d}_\theta(x_j \to S_i) \le \tau_i$ and $\widetilde{d}_\theta(x_i \to S_j) \le \tau_j$ does not add any cross-component edge.*

3. ***(Lower/bridge)** If*

$$\frac{\Delta}{h_{\max}} < \underline{C} - \kappa_- \frac{\sigma}{h_{\max}}, \qquad \text{and} \qquad \Delta - 2\sigma \le \min\left\{\frac{r_*}{a}, \frac{r_*}{1+2a}\right\} \tag{11}$$

   *(for some fixed $a \in (0, 1/8)$; the choice $1/8$ is convenient because $(1 + 2a) \le 5/4$), then a bridging edge appears in the union-kNN graph with probability at least*

$$1 - 2\exp\left(-\eta\,n_{\min}(\Delta - 2\sigma)^d\right) - \exp(-\gamma k),$$

   *where $\eta = \underline{c}\,a^d$ and $\gamma > 0$ are absolute constants.*

*Proof.* Intersect the following events, each holding with probability $\ge 1 - \delta/5$ after adjusting constants: (i) the noiseless fill-distance sandwich (Lemma A.1); (ii) the uniform $k$NN bound (Lemma A.2); (iii) the noisy sandwich (Lemma A.10); (iv) the two-scale bounds (Lemmas A.11–A.13).

For (1), any cross pair $(i, j)$ satisfies

$$\|x_i - x_j\| \ge \|\pi_{\mathcal{M}}(x_i) - \pi_{\mathcal{M}}(x_j)\| - \|\xi_i\| - \|\xi_j\| \ge \Delta - 2\sigma.$$

On the other hand, by Lemmas A.2 and A.10,

$$\sqrt{H_i H_j} \le \max\{H_i, H_j\} \le \overline{C}\,h_{\max} + C_2\sigma.$$

Thus if $\Delta - 2\sigma > \overline{C}\,h_{\max} + C_2\sigma$, i.e. $\Delta/h_{\max} > \overline{C} + (2 + C_2)\,\sigma/h_{\max}$, the Euclidean gate removes $\{i, j\}$; triangle support cannot revive it. This yields equation 10 with $\kappa_+ = 2 + C_2$.

For (2), consider $y_{i \leftarrow j} := \widetilde{d}_\theta(x_j \to S_i)/H_i$. By Lemma A.12,

$$y_{i \leftarrow j} \ge \frac{B_0'(\Delta - 2\sigma)}{H_i} - \frac{A_0}{H_i}\left(\frac{k}{n_s}\right)^{1/d} h_s - \frac{C_\sigma\,\sigma}{H_i}.$$

Lemma A.10 gives $H_i \ge c_0\,h_{\max}$ for some $c_0 \in (0, 1)$ (depending on $c_{\text{noise}}$), hence

$$y_{i \leftarrow j} \ge \frac{B_0'}{c_0} \cdot \frac{\Delta}{h_{\max}} - C' \cdot \frac{\sigma}{h_{\max}} - C''\left(\frac{k}{n_s}\right)^{1/d} \frac{h_s}{h_{\max}}.$$

By Lemma A.11, the same-component quantile $\tau_i$ obeys $\tau_i \le C'''(k/n_s)^{1/d}(h_s/H_i) + o(1) \le C''''(h_s/h_{\max}) + o(1)$. Under equation 10, for $\kappa_+$ large enough to absorb these terms, one has $y_{i \leftarrow j} > \tau_i$. The same bound holds from $j$'s side, so the add-only rule does not add any cross-component edge.

For (3), apply the noiseless bridging proof (Proposition A.4) with $\Delta$ replaced by $\Delta_{\mathrm{eff}} := \Delta - 2\sigma$. Choose intrinsic caps of radii $a\Delta_{\mathrm{eff}}$ and use radius $\rho = B(a)\Delta_{\mathrm{eff}}$ for the same-component crowding test. The small-radius condition in equation 11 ensures both radii lie within the bi-Lipschitz regime. Exactly as in the noiseless case,

$$\mathbb{E}[S_x] \leq (n_1 - 1)\,\overline{c}\,B(a)^d\,\Delta_{\mathrm{eff}}^d \leq n_1\,\overline{c}\,B(a)^d\left(\underline{C} - \kappa_-\tfrac{\sigma}{h_{\max}}\right)^d h_{\max}^d.$$

If $\Delta/h_{\max} < \underline{C} - \kappa_-\sigma/h_{\max}$ with $\kappa_-$ chosen to compensate for the $O(\sigma)$ slack in Lemma A.10, then $\mathbb{E}[S_x] \leq k/4$, whence $\Pr(S_x \geq k/2) \leq e^{-\gamma k}$. Cap-occupancy holds with probability at least $1 - 2\exp(-\eta n_{\min}\Delta_{\mathrm{eff}}^d)$, producing a cross edge with the stated probability. $\qquad\square$

### A.3.5 Adaptive Local Fill Distance and the Floor-Anchored Local-$k$ Schedule

**Why an adaptive fill proxy?** A single global degree $k$ produces $k$NN radii $D_k(x)$ that fluctuate with local sampling density: dense regions yield tiny radii, sparse regions yield large ones. This heterogeneity harms both (i) the *geometric-mean* Euclidean gate $\|x_i - x_j\| \leq \sqrt{H_i H_j}$ (decisions become asymmetric when one endpoint is much denser) and (ii) the add-only rescue, whose directional statistic normalizes by a per-node scale. Our remedy is to *equalize* the intrinsic neighborhood scale by adapting $k$ per node, while *never* dropping below the RGG-safe pilot $k^\star$.

**Setup and notation.** Let $\mathcal{M} \subset \mathbb{R}^D$ be a finite union of compact, connected $d$-dimensional $C^2$ submanifolds with positive reach, and assume the two-sided local mass bounds on a fixed small-ball scale $r_* > 0$: there exist $0 < \underline{c} \leq \overline{c} < \infty$ such that for all $x \in \mathcal{M}$, $0 < r \leq r_*$,

$$\underline{c}\,r^d \leq \mu(B(x,r)) \leq \overline{c}\,r^d,$$

where $\mu$ is the (componentwise) normalized surface measure. For a node $x$, write $D_k(x)$ for the $k$NN radius. We denote the (unknown) clean fill distance by $h$ and use $H$ for computable per-node proxies.

### A.4 The adaptive fill-distance proxy

**Pilot radii and local degrees.** We work at a connectivity-safe pilot $k^\star = \lceil \log(4n/\delta) \rceil \in \{2, \ldots, n-1\}$ and compute *pilot* radii $H_i^{\mathrm{pilot}} := D_{k^\star}(x_i)$. Let $H_{\mathrm{ref}} := \mathrm{median}\{H_i^{\mathrm{pilot}} > 0\}$ and choose

$$k_{\min} := \left\lceil 0.5 \log\frac{4n}{\delta} \right\rceil, \qquad k_{\max} := \min\{n-1, 3k^\star\}.$$

We then set the per-node degree by

$$k_i = \max\left(k^\star,\ \mathrm{clip}\big(\lfloor k^\star\,(H_{\mathrm{ref}}/\max(H_i^{\mathrm{pilot}}, 10^{-12}))\,^{d_{\mathrm{eff}}}\rfloor,\ k_{\min},\ k_{\max}\big)\right), \tag{12}$$

and define the *local fill proxy* $H_i := D_{k_i}(x_i)$. The geometric-mean gate and all normalizations use $H_i$.

**Why equation 12 equalizes scale.** Under the mass bounds, standard order-statistic arguments imply

$$D_k(x) = \Theta\left(\left(\tfrac{k}{n}\right)^{1/d}\right) \qquad \text{(clean, uniformly in } x\text{)}. \tag{13}$$

More precisely, with probability $\geq 1 - \delta$, there exist explicit $C_-, C_+ > 0$ depending only on $(d, \underline{c}, \overline{c})$ such that

$$C_-\left(\tfrac{k}{n}\right)^{1/d} \leq D_k(x) \leq C_+\left(\tfrac{k}{n}\right)^{1/d} \quad \forall x \in \mathcal{M},\ \forall k \in [k_{\min}, k_{\max}]. \tag{14}$$

Heuristically, $D_k(x) \approx \left(\tfrac{k}{n\,c(x)}\right)^{1/d}$, where $c(x) \in [\underline{c}, \overline{c}]$ is the local mass constant. Evaluated at the pilot, $H_i^{\mathrm{pilot}} \approx \left(\tfrac{k^\star}{n\,c(x_i)}\right)^{1/d}$, so $c(x_i) \approx \tfrac{k^\star}{n}\,(H_i^{\mathrm{pilot}})^{-d}$. To make $D_{k_i}(x_i)$ *match a target radius* $r_{\mathrm{tgt}}$, we would set $k_i \approx n\,c(x_i)\,r_{\mathrm{tgt}}^d$. Plugging the pilot estimate of $c(x_i)$ and choosing $r_{\mathrm{tgt}} := H_{\mathrm{ref}}$ gives $k_i \approx k^\star\,(H_{\mathrm{ref}}/H_i^{\mathrm{pilot}})^d$, which is equation 12 with $d_{\mathrm{eff}}$ in place of $d$ and with clipping/flooring for stability.

---

**Algorithm 2** MBC: Euclidean Gate, Triangle Support, Quantile Two-Scale DTM Rescue

---

**Require:** $X \in \mathbb{R}^{n \times D}$, $\delta, \alpha \in (0, 1)$

1: **Fixed:** $\theta \leftarrow 2$, $q_\tau \leftarrow 0.90$, $c_{\text{trim}} \leftarrow 4$, $S_{\max} \leftarrow 32$, $t_\triangle \leftarrow 2$
2: **Standardize** $X$; set $d_{\text{eff}} \leftarrow$ #PCA comps for $\geq 90\%$ EVR (cap 64); $k^\star \leftarrow \lceil \log(4n/\delta) \rceil$
3: **Pilot** $k^\star$: get $H_i^{\text{pilot}} = D_{k^\star}(x_i)$; set $H_{\text{ref}} \leftarrow \text{median}\{H_i^{\text{pilot}}>0\}$, $k_{\min} \leftarrow \lceil 0.5 \log(4n/\delta) \rceil$, $k_{\max} \leftarrow \min(n-1, 3k^\star)$
4: **Local-$k$:** $k_i \leftarrow \max\big(k^\star,\ \text{clip}(\lfloor k^\star (H_{\text{ref}}/\max(H_i^{\text{pilot}}, 10^{-12}))^{d_{\text{eff}}} \rfloor,\ k_{\min}, k_{\max})\big)$
5: **$k$NN & candidates:** for each $i$, get $N_i$ (top-$k_i$) and $H_i = D_{k_i}(x_i)$; $P = \{\{i,j\}: j \in N_i \text{ or } i \in N_j\}$
6: **Euclidean gate:** $E_{\text{eucl}} \leftarrow \{\{i,j\} \in P :\ \|x_i - x_j\| \leq \sqrt{H_i H_j}\}$
7: **Triangle support:** $E_{\text{tri}} \leftarrow \{\{i,j\} \in E_{\text{eucl}} :\ |N_i \cap N_j| \geq t_\triangle\}$
8: **Rescue-eligible:** $R \leftarrow E_{\text{eucl}} \setminus E_{\text{tri}}$
9: **for** $i = 1$ to $n$ **do**                                                                                  ▷ per-node $\tau_i$
10:     $S_i \leftarrow \{q \in N_i :\ \|x_q - x_i\| \leq c_{\text{trim}} H_i\}$; if $|S_i| > S_{\max}$, keep closest $S_{\max}$
11:     $z_q \leftarrow \text{TwoScaleDTM}(x_q \mid S_i, \theta)/H_i$;   $\tau_i \leftarrow \text{Quantile}_{q_\tau}\{z_q : q \in S_i\}$
12: **end for**
13: **for** each $\{i,j\} \in R$ **do**                                                                           ▷ add-only rescue
14:     $y_{i \leftarrow j} \leftarrow \text{TwoScaleDTM}(x_j \mid S_i, \theta)/H_i$; $y_{j \leftarrow i} \leftarrow \text{TwoScaleDTM}(x_i \mid S_j, \theta)/H_j$
15:     **if** $y_{i \leftarrow j} \leq \tau_i$ and $y_{j \leftarrow i} \leq \tau_j$ **then**
16:         $E_{\text{tri}} \leftarrow E_{\text{tri}} \cup \{\{i,j\}\}$
17:     **end if**
18: **end for**
19: **Clusters:** labels $L \leftarrow \text{CC}(V = [n], E_{\text{tri}})$
20: **$K$-bracket (remove-only):** $\varepsilon_k \leftarrow \sqrt{\frac{1}{2}\frac{\log(2n/\alpha)}{k^\star}}$ (clip $\leq 0.45$);   recompute GM + Triangle at scales $(1 \pm \varepsilon_k) \cdot k_i$ to get $K_{\text{CI}} = [K(1+\varepsilon_k), K(1-\varepsilon_k)]$
21: **N1 noise:** on the $(1+\varepsilon_k)$ remove-only graph, mark nodes with degree $\leq 1$ as noise ($L_i \leftarrow -1$)

---

## B  Empirical Analysis

**Algorithm Pseudocode**   We detail the pseudocode for the MBC algorithm below:

**Default Hyperparameters (All Experiments)**   Unless otherwise noted, all results use a single, dataset-agnostic configuration with no per-dataset tuning. Table 2 lists the exposed knobs and fixed design choices; these settings were held constant across all benchmarks.

**Notes.**   (*i*) **Triangle support** $= 2$ is the default; $= 1$ is too permissive (admits bridges), while $= 3$ can over-fragment the remove-only bracket on sparse scales. (*ii*) **DTM rescue** is conservative: it often does not fire on clearly separable data; disabling it in that regime yields $2 \sim 3 \times$ faster runs without changing ARI/NMI. (*iii*) The $K$-bracket is a *monotonicity diagnostic* from remove-only graphs; large widths typically reflect micro-fragmentation at sparser scales rather than errors in the final labels. (*iiii*) Additional ablations (triangle strength, DTM thresholds, PCA EVR) were done on our algorithm using the datasets in Table 4 and showed our default parameters are robust on clean/separable regimes. That is, our choice in such parameters, including $q_{high}$ and $\theta$, did not noticeably change our decided $K$ nor bracket for reasonable perturbations.

### B.0.1  Extended Results (Noise and Anisotropy)

**How the Baselines Are Configured**   Let each method be run with library defaults to reflect typical out-of-the-box usage. We specify the details in the below table.

**Hyperparameter sweeps on neural datasets**   To assess how sensitive standard clustering algorithms are to hyperparameter choice on the neural datasets, we sweep a grid of settings for each method and summarize the resulting distribution of cluster counts $K$ in the table below.

| Group | Name (symbol) | Default and rationale |
|---|---|---|
| Pilot & bracket | Failure budget ($\delta$) | 0.05 (sets pilot scale $k^\star = \lceil \log(4n/\delta) \rceil$) |
| | Bracket level ($\alpha_{\text{CI}}$) | 0.05 (defines $\varepsilon_k = \sqrt{\frac{1}{2}\log(2n/\alpha_{\text{CI}})/k^\star}$; clip 0.45) |
| | $k$ bracket | Report $[K(1+\varepsilon_k),\ K(1-\varepsilon_k)]$ on *remove-only* graphs |
| Graph construction | Local-$k$ schedule | Floor-anchored: $k_i = \max\big(k^\star,\ \lfloor k^\star (\text{med}(H)/H_i)^{d_{\text{eff}}} \rfloor\big)$, with $k_{\min} = \lceil 0.5\log(4n/\delta) \rceil$, $k_{\max} = 3k^\star$ |
| | Candidate edges | *Union-k*: keep $\{i,j\}$ if $i \in N_j$ or $j \in N_i$ |
| | Euclidean gate | Keep $\{i,j\}$ if $\|x_i - x_j\| \leq \sqrt{H_i H_j}$ (local, scale-adaptive) |
| | Triangle support | Require $|N_i \cap N_j| \geq \mathbf{2}$ (suppresses one-sided coincidences) |
| DTM rescue (add-only) | Enabled | **True** by default (off in certain ablations experiments) |
| | Two-scale factor ($\theta$) | 2.0 (radius-doubling statistic; stable and simple) |
| | Quantile ($q_{\text{high}}$) | 0.90 (mutual typicality threshold; conservative) |
| | Trimming / cap | Multiplier $c{=}4$, within-set cap $S_{\max}{=}32$ |
| Representation | Standardization | z-score per feature |
| | $d_{\text{eff}}$ (PCA) | Smallest #PCs for $\geq \mathbf{90}\%$ explained variance (cap 64) |
| | Tangent projection | Run MBC on PCA scores |
| Noise handling | N1 heuristic | On bracket-high remove-only graph, mark degree $\leq 1$ as noise |
| | $K$ reporting | $K$ counts all labels including $-1$; we also report the $K$-bracket from remove-only graphs |

Table 2: MBC defaults used in all experiments. No per-dataset tuning.

Table 3: **Baseline configuration**

| Method | Library | Defaults and optional sweep |
|---|---|---|
| DBSCAN | scikit-learn | *Default:* `eps`=0.5, `min_samples`=5 (Euclidean). |
| OPTICS | scikit-learn | *Default:* Euclidean metric; `min_samples`=5; `xi`=0.05. |
| BIRCH | scikit-learn | *Default:* threshold $= 0.5$; branching factor $= 50$; $n_{\text{clusters}} = $ None. *Parameter sweep (For appendix table only):* threshold $\in \{0.3, 0.5, 0.7\}$; branching factor $\in \{25, 50, 100\}$. |
| HDBSCAN | `hdbscan` | *Default:* `min_cluster_size` $= \max\{5, \lfloor 0.02n \rfloor\}$; `min_samples`=None; Euclidean metric; cluster selection=`leaf`. |

Table 4: **Extended results** (three seeds; best per row in **bold**). "MBC Bracket" is the median across runs of the monotone component-count interval.

| Dataset | Method | ARI ↑ | NMI ↑ | Mean $K$ | MBC Bracket |
|---------|--------|-------|-------|----------|-------------|
| **Two Moons** 
 *2D; additive Gaussian noise* 0.08; $K_{true}$=2 | MBC | 0.333 | 0.333 | 1.33 | [1, 8] |
| | OPTICS | 0.006 | 0.182 | 112.00 | – |
| | BIRCH | **0.558** | **0.577** | 3.00 | – |
| | HDBSCAN | 0.083 | 0.254 | 7.33 | – |
| **Concentric Circles** 
 *2D; factor* 0.3, *noise* 0.06; $K_{true}$=2 | MBC | **1.000** | **0.999** | 2.33 | [2, 8] |
| | OPTICS | 0.007 | 0.186 | 119.67 | – |
| | BIRCH | 0.250 | 0.347 | 3.00 | – |
| | HDBSCAN | 0.038 | 0.232 | 10.67 | – |
| **Gaussian Blobs** 
 *10D; std* 3.0; $K_{true}$=6 | MBC | 0.465 | 0.583 | 4.33 | [3, 25] |
| | OPTICS | 0.096 | 0.274 | 4.67 | – |
| | BIRCH | **0.509** | **0.734** | 3.00 | – |
| | HDBSCAN | 0.439 | 0.628 | 7.00 | – |
| **Gaussian Blobs** 
 *25D; std* 3.5; $K_{true}$=6 | MBC | **0.856** | **0.938** | 6.67 | [6, 17] |
| | OPTICS | 0.716 | 0.857 | 5.67 | – |
| | BIRCH | 0.509 | 0.734 | 3.00 | – |
| | HDBSCAN | 0.783 | 0.857 | 7.00 | – |
| **Gaussian Blobs (Anisotropic)** 
 *20D;* $K_{true}$=6 | MBC | **0.998** | **0.997** | 7.67 | [6, 18] |
| | OPTICS | 0.856 | 0.936 | 6.67 | – |
| | BIRCH | 0.478 | 0.722 | 3.00 | – |
| | HDBSCAN | 0.994 | 0.993 | 6.67 | – |
| **Gaussian Blobs (Variable Variance)** 
 *2D;* $K_{true}$=3 | MBC | 0.381 | 0.489 | 2.00 | [2, 11] |
| | OPTICS | 0.006 | 0.243 | 119.00 | – |
| | BIRCH | 0.495 | 0.590 | 3.00 | – |
| | HDBSCAN | **0.784** | **0.816** | 3.67 | – |
| **Gaussian Blobs** 
 *2D; std* 0.9; $K_{true}$=4 | MBC | 0.388 | 0.607 | 2.33 | [2, 7] |
| | OPTICS | 0.006 | 0.278 | 111.00 | – |
| | BIRCH | **0.653** | **0.776** | 3.00 | – |
| | HDBSCAN | 0.756 | 0.816 | 4.67 | – |
| **Gaussian Blobs (Anisotropic)** 
 *2D;* $K_{true}$=4 | MBC | 0.111 | 0.188 | 2.33 | [1, 34] |
| | OPTICS | 0.008 | 0.294 | 115.33 | – |
| | BIRCH | **0.675** | **0.793** | 3.00 | – |
| | HDBSCAN | 0.504 | 0.636 | 6.33 | – |
| **Gaussian Blobs** 
 *3D; std* 2.3; $K_{true}$=5 | MBC | 0.070 | 0.157 | 1.67 | [1, 6] |
| | OPTICS | 0.004 | 0.227 | 64.33 | – |
| | BIRCH | **0.555** | **0.741** | 3.00 | – |
| | HDBSCAN | 0.390 | 0.596 | 5.67 | – |
| **Fashion–MNIST** 
 *28×28 grayscale; PCA→50;* $K_{true}$=10 | MBC | 0.000 | 0.002 | 3.00 | [10, 30] |
| | OPTICS | 0.000 | 0.041 | 29.00 | – |
| | BIRCH | **0.124** | **0.307** | 3.00 | – |
| | HDBSCAN | 0.000 | 0.000 | 1.00 | – |
| **Wine** 
 *13 features (tabular);* $K_{true}$=3 | MBC | 0.000 | 0.000 | 1.00 | [1, 5] |
| | OPTICS | 0.036 | 0.195 | 5.00 | – |
| | BIRCH | **0.790** | **0.786** | 3.00 | – |
| | HDBSCAN | 0.266 | 0.361 | 3.00 | – |
| **Breast Cancer** 
 *30 features (tabular);* $K_{true}$=2 | MBC | 0.007 | 0.015 | 2.00 | [2,...**4**,...9] |
| | BIRCH | **0.536** | **0.443** | 3.00 | – |
| | HDBSCAN | 0.000 | 0.000 | 1.00 | – |

Table 5: **Neural data: cluster-count variability across hyperparameters.** For each dataset and method we sweep a grid of hyperparameters and summarize the resulting distribution of cluster counts $K$. $K_{\min}$, $K_{\max}$, and $K_{\mean}$ are the minimum, maximum, and mean over the grid; "% within $\pm 2$" and "% exact" denote the fraction of runs whose $K$ lies in $\{K_{\text{true}} - 2, \ldots, K_{\text{true}} + 2\}$ or equals $K_{\text{true}}$. Across the three neural datasets we evaluate 144 HDBSCAN configurations (`min_cluster_size` × `min_samples`), 360 DBSCAN configurations (`eps` × `min_samples`), 134 K-Means configurations (`n_clusters`), 134 GMM configurations (`n_components`), and 31 Spectral configurations (`n_clusters`). For K-Means and Spectral Clustering, we vary the number of clusters from the ranges as described, omitting clusters with less than 2 percent of the present points, denoting them as noise.

| Dataset | Method | $K_{\text{true}}$ | $K_{\min}$ | $K_{\max}$ | $K_{\mean}$ | % within $\pm 2$ | % exact |
|---|---|---|---|---|---|---|---|
| | DBSCAN | 1 | 0 | 36 | 1.9 | 90.8 | 48.3 |
| | GMM | 1 | 1 | 50 | 25.5 | 6.0 | 2.0 |
| **V1** | HDBSCAN | 1 | 0 | 35 | 2.9 | 87.5 | 0.0 |
| | K-Means | 1 | 1 | 50 | 25.5 | 6.0 | 2.0 |
| | Spectral | 1 | 2 | 20 | 11.0 | 10.5 | 0.0 |
| | DBSCAN | 8 | 1 | 66 | 4.7 | 8.3 | 3.3 |
| | GMM | 8 | 1 | 50 | 25.5 | 10.0 | 2.0 |
| **Retina (labeled only)** | HDBSCAN | 8 | 5 | 92 | 12.2 | 64.6 | 6.2 |
| | K-Means | 8 | 1 | 50 | 25.5 | 10.0 | 2.0 |

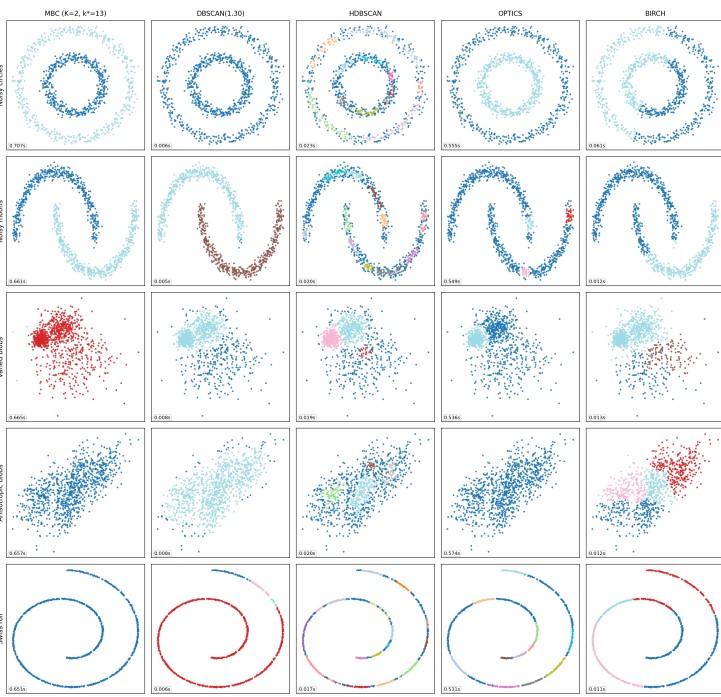

Figure 3: A visual summary of performance on canonical clustering datasets for MBC (left column) against current state-of-the-art algorithms (DBSCAN, HDBSCAN, OPTICS and BIRCH) with default parameters (Table B.0.1).

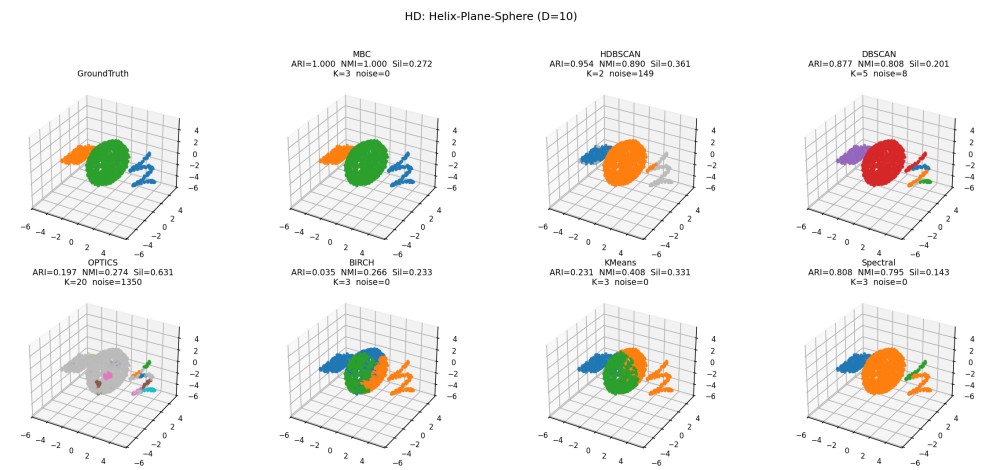

Figure 4: **Mixed-dimension, high-D stress test (Helix–Plane–Sphere, $D=10$).** We synthesize three manifold pieces with different intrinsic dimensions in $\mathbb{R}^3$—a 1D helix, a 2D plane patch, and a noisy 3D sphere—then embed them into $\mathbb{R}^{10}$ via a random orthonormal map and add small isotropic tubular noise. Each panel shows the predicted partition together with ARI, NMI, silhouette score (Sil) (**?**), the number of clusters $K$, and the count of points labeled as "noise" by the method. *MBC* recovers all three components exactly despite the heterogeneous shapes and dimensions. Density/graph baselines either merge or overfragment components and often declare large fractions of points as noise (e.g., OPTICS, DBSCAN); *HDBSCAN* collapses two structures ($K = 2$). Centroid/spectral (K-Means, Spectral) methods given $K_{\text{true}}$ fail due to these being nonconvex, anisotropic manifolds.

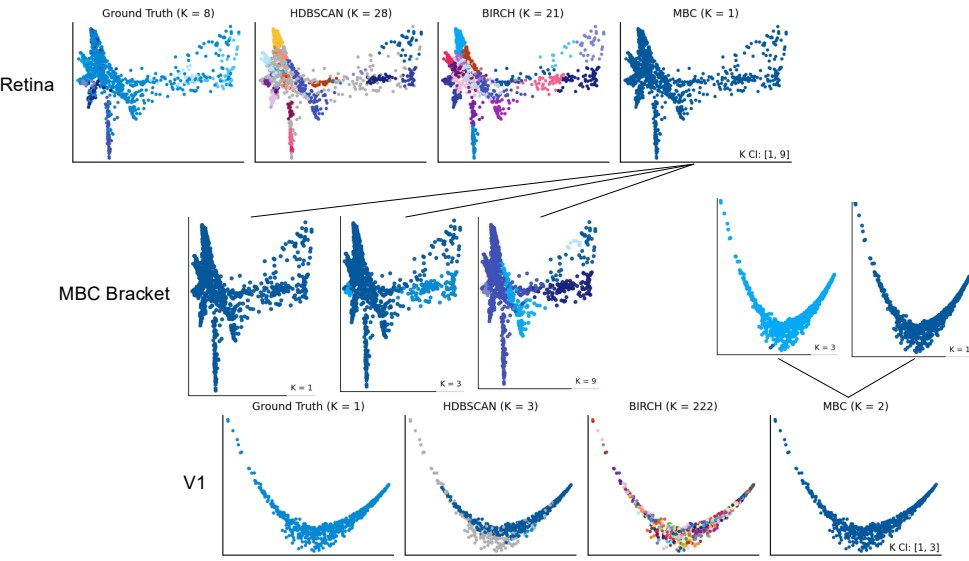

Figure 5: **Retina vs. V1—MBC bracket reveals sampling-limited ambiguity.** Top row: Retina—ground truth ($K=8$) vs. HDBSCAN ($K=28$), BIRCH ($K=21$), and an MBC partition (point $K=1$) with reported $K_{\text{CI}}=[1,9]$. Middle: *MBC bracket panels* at $K \in \{1,3,9\}$ illustrate the plausible range supported by the data. Bottom row: V1—ground truth ($K=1$) vs. HDBSCAN ($K=3$), BIRCH ($K=222$), and an MBC partition (point $K=1$) with bracket $[1,3]$. MBC refrains from forcing clusters and instead reveals the intrinsic transitional regime via the bracket.

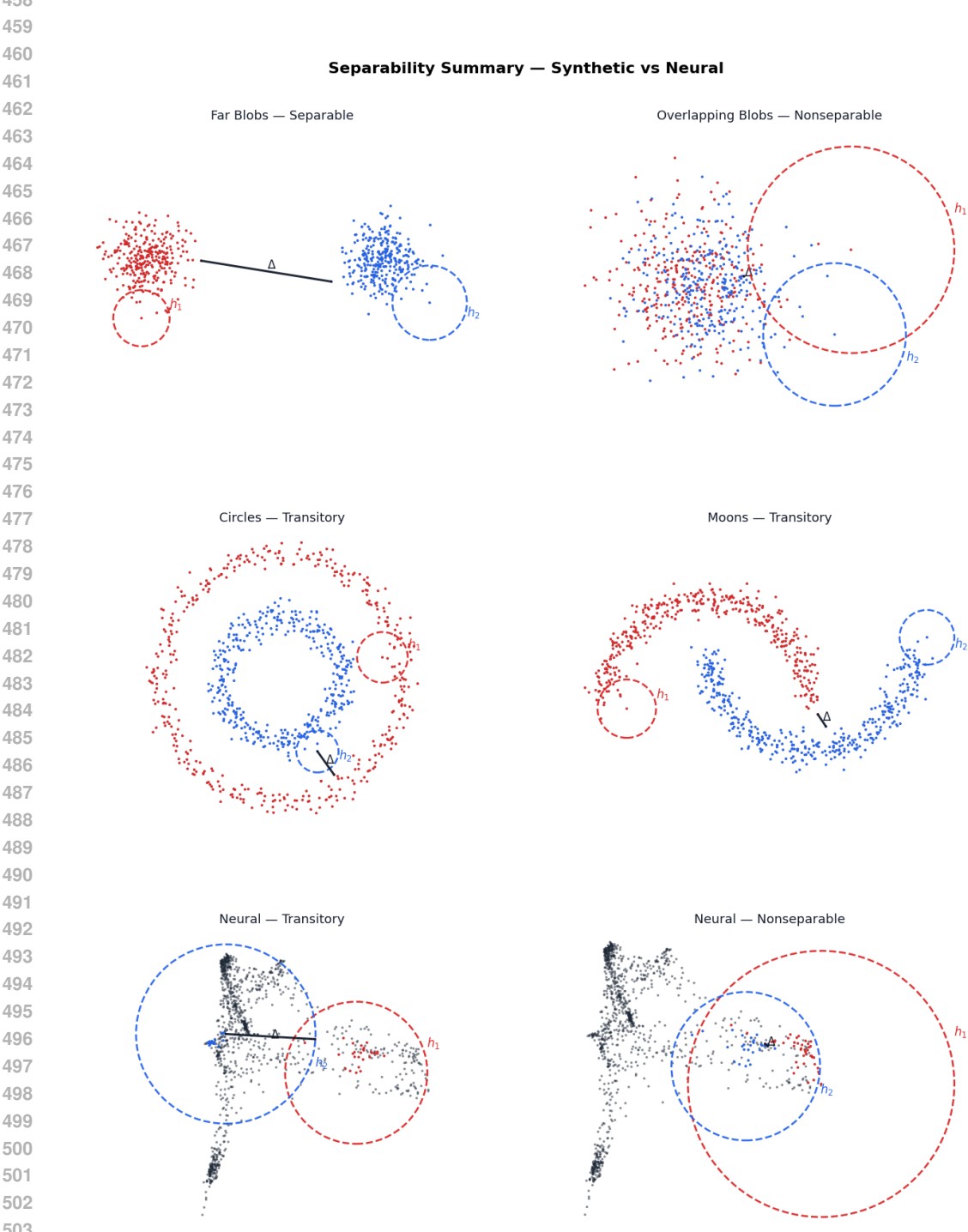

Figure 6: **Separability summary—synthetic and neural.** Each panel shows two components with ambient offset $\Delta$ and worst-case fill radii $h_1, h_2$ (dashed). Top: well-separated and overlapping Gaussian blobs (separable vs. nonseparable). Middle: *Circles* and *Moons* at intermediate noise (transitional). Bottom: retinal neuron pairs exhibit both transitional and nonseparable cases. Larger $\Delta/h_{\max}$ favors separability; small $\Delta/h_{\max}$ induces bridging and fusion.

