# OpenReview forum: "One cluster or two? A Manifold-Based Approach"
_ICLR.cc/2026/Conference — Submitted to ICLR 2026_

### Official Review · Reviewer_gyYv · 2025-10-29

**Soundness:** 2
**Presentation:** 2
**Contribution:** 2
**Rating:** 2
**Confidence:** 4

**Summary:**

The paper presents a method for determining whether samples are "truly" from separate manifold components (under the manifold hypothesis). It then presents a way to cluster based on this notion and evaluates this clustering against density-based methods on toy datasets.

**Strengths:**

The paper proposes a novel way to make clustering and high-dimensional point evaluations based on the manifold hypothesis. I appreciate the efforts to formalize elements of the manifold hypothesis and to transfer this formalization into a practical set of decision-making criteria. I also find that the exposition and motivation are well-structured and that the algorithmic choices are well motivated. In short, I commend the paper's goal and believe the authors approached this goal in an appropriate way.

**Weaknesses:**

Unfortunately, I also think the authors are too focused on the specifics of making things sound nice and well-motivated. This comes at the expense of a practically sound and effective method. This is evidenced first in the experimental results. Specifically, the proposed algorithm only works in toy settings where the separation between clusters is extremely clear. In these settings, there is a large number of algorithms which will find identical clusterings (single-linkage clustering, ward algorithm, spectral clustering, etc.). Indeed, showing effectiveness on 2-moons and concentric circles is a common first step when showing experimentally that a non-convex clustering works as intended.

However, the authors' chosen algorithm immediately stops working once we reach "real" data. I would note that these real datasets are still fairly simple and much cleaner than the *actual* messiness of large, high-dimensional datasets. Indeed, I am confident that an effective method would be to run UMAP on the high-dimensional datasets and then apply HDBSCAN to the outputs. The effectiveness of these standard clustering procedures suggests that there *is* structure to cluster along.

Nonetheless, the authors' proposed algorithm stops working effectively on these real-world datasets. The authors suggest that this is because, for real datasets, the assumption of multiple distinct manifolds is broken. I agree with this conjecture, but I also argue that this significantly weakens the authors' proposed algorithm. Namely, *all* reasonably sophisticated real-world datasets will violate the assumptions of having multiple distinct, well-separated and well-sampled manifolds. Consequently, the authors' theoretical analysis, while interesting, does not naturally transfer to real-world settings.

What I would find more compelling is to find an analysis which explains why, for example, doing tSNE/UMAP into density-based clustering (or spectral, single-linkage, etc.) works as well as it does. It seems the authors' work could be naturally extended to such an analysis, and this would make the work more practically valuable.

**Questions:**

These are smaller questions than those implied in the weaknesses section:
- Why is the first theorem statement so long? Ideally, the assumptions/requirements should be stated outside the theorem environment so that the theorem statement itself can be made simple to interpret.
- Why do the authors state that their clustering algorithm does not require hyperparameters? It seems that, later on that page, they specify several hyperparameters which they've chosen.
- Some words are used without being introduced. For example, I am not familiar with the concept of "tubular" noise. Similarly, I'm not sure that manifolds are ever formally introduced. In general, although I could follow the statements made in the paper, I fear that somebody outside of our narrow subfield would struggle interpreting the statements.
- Is it clear why we are applying random geometric graph asymptotics to the analysis? The assumption that the samples exist on manifolds with varying density seems to contradict the fact that these are random geometric graphs. Consequently, none of the connectivity properties of random graphs should apply.

---

> ### Author Response · Authors · 2025-11-23
>
> We thank the reviewer for the thoughtful, candid, and highly constructive feedback. We especially appreciate the recognition that the paper attempts to operationalize the manifold hypothesis by formalizing its elements into a practical set of decision-making criteria, and that the exposition, motivation, and algorithmic choices are, in spirit, well structured and appropriately targeted to that goal. This captures the intent behind our geometric framing and the bracket construction.
>
> In the revised manuscript, we have incorporated a substantial set of improvements addressing the reviewer’s insights, as listed below:
>
> - “Unfortunately, I also think the authors are too focused on the specifics of making things sound nice and well-motivated. This comes at the expense of a practically sound and effective method. This is evidenced first in the experimental results. Specifically, the proposed algorithm only works in toy settings where the separation between clusters is extremely clear. In these settings, there is a large number of algorithms which will find identical clusterings (single-linkage clustering, ward algorithm, spectral clustering, etc.).”
>
> In the revision we have reframed the synthetic examples as sanity checks for the offset-to-fill-distance picture, and made explicit that MBC is not proposed as a superior optimizer on Two Moons / Circles but as a geometric test of clusterability and sampling scale. The main empirical emphasis is now on settings like the retina and V1 recordings, where the point is precisely that many standard methods can produce attractive partitions while the underlying manifold geometry and sampling density do not clearly support a unique clustering.
>
> - “However, the authors' chosen algorithm immediately stops working once we reach "real" data. I would note that these real datasets are still fairly simple and much cleaner than the actual messiness of large, high-dimensional datasets. Indeed, I am confident that an effective method would be to run UMAP on the high-dimensional datasets and then apply HDBSCAN to the outputs. The effectiveness of these standard clustering procedures suggests that there is structure to cluster along.”
>
> We’d like to repeat that a key message in the paper is that, for many real world problems, whether clusters exist is fundamentally difficult to determine. The motivation provided by putting the neuroscience task at the Introduction is to emphasize how dangerous it can be to apply a technique (or multiple techniques) to an unknown problem. __How to interpret these apparent failures is the core signal of the method__: on Digits, MNIST, and especially the full retina dataset, MBC returns a wide bracket and low agreement with labels, indicating that at the given representation and sampling density the data sit in a transitory or nonseparable regime rather than supporting a confident partition. In contrast, pipelines such as UMAP+HDBSCAN are designed to extract structure even in this regime; we now position MBC as complementary—once an embedding is chosen (i.e. UMAP or t-SNE), MBC can be applied as a scale-aware diagnostic of whether the resulting clusters are reasonable at that embedding and density.
>
> - “Nonetheless, the authors' proposed algorithm stops working effectively on these real-world datasets. The authors suggest that this is because, for real datasets, the assumption of multiple distinct manifolds is broken. I agree with this conjecture, but I also argue that this significantly weakens the authors' proposed algorithm. Namely, all reasonably sophisticated real-world datasets will violate the assumptions of having multiple distinct, well-separated and well-sampled manifolds. Consequently, the authors' theoretical analysis, while interesting, does not naturally transfer to real-world settings.What I would find more compelling is to find an analysis which explains why, for example, doing tSNE/UMAP into density-based clustering (or spectral, single-linkage, etc.) works as well as it does. It seems the authors' work could be naturally extended to such an analysis, and this would make the work more practically valuable.”
>
> We agree that the idealized manifold assumptions are rarely met exactly, and in the revision we highlight this in the introduction. Regarding limitations: we emphasize that the bracket is intended to quantify how far a given dataset and representation are from the “clean regime” rather than to claim that all real data satisfy it. Nevertheless, we appreciate the reviewer’s point, so we also now explicitly discuss, in the discussion section, how the offset–fill-distance framework could be extended to analyze multi-stage pipelines (e.g., UMAP or spectral embeddings followed by density-based clustering), and we view such an analysis as a natural direction for future work rather than something we can fully develop within the space of this paper.

---

> ### Author Response · Authors · 2025-11-23
>
> Below, we respond to the specific questions raised:
>
> - “Why is the first theorem statement so long? Ideally, the assumptions/requirements should be stated outside the theorem environment so that the theorem statement itself can be made simple to interpret.”
>
> You’re right; and we take your point through much of the paper. Specifically,, we have reorganized Theorem 1 so that the sampling model, geometric assumptions, and graph construction appear as labeled subheads preceding the main “threshold statement,” which is now focused on the two regimes (no cross edges vs. bridging) and a separate remark explaining the behavior and interpretation of the constants. This keeps the theorem’s logical content intact while making the core message—the emergence of a separation threshold–is easier to parse.
>
> - “Why do the authors state that their clustering algorithm does not require hyperparameters? It seems that, later on that page, they specify several hyperparameters which they've chosen.”
>
> Again, you’re right. We have softened the language to describe MBC as “parameter-light” rather than “hyperparameter-free,” and we now clearly distinguish between the scale parameter k^*, which is dictated by the connectivity theory, and the rescue and bracket constants, which are fixed once and used unchanged across all experiments. The added corollary and sensitivity discussion clarify that these constants primarily control bracket width in ambiguous regimes and have little effect when separation is clearly favorable or clearly absent.
>
> - “Some words are used without being introduced. For example, I am not familiar with the concept of "tubular" noise. Similarly, I'm not sure that manifolds are ever formally introduced. In general, although I could follow the statements made in the paper, I fear that somebody outside of our narrow subfield would struggle interpreting the statements.”
>
> We have added a dedicated background section that introduces manifolds, reach, tubular neighborhoods, and the tubular-noise model with standard references in the appendix.
>
> - “Is it clear why we are applying random geometric graph asymptotics to the analysis? The assumption that the samples exist on manifolds with varying density seems to contradict the fact that these are random geometric graphs. Consequently, none of the connectivity properties of random graphs should apply.”
>
> We understand the source of confusion and have clarified this point in the revised background and theorem statements. Classical random geometric graph (RGG) asymptotics are often presented for uniform samples in Euclidean boxes, but what they require are local structure, not global uniformity. Following the manifold reconstruction framework of Niyogi, P., Smale, S. & Weinberger, S. Finding the Homology of Submanifolds with High Confidence from Random Samples. Discrete Comput Geom 39, 419–441 (2008), we assume each component is a compact $C^2$ submanifold with positive reach and that the sampling measure has a density bounded above and below with respect to surface measure, so that for all sufficiently small radii $r$, we can bound the mass of balls with our two-sided volume condition in Theorem 1. This condition captures “varying density” at larger scales while ensuring that, in small neighborhoods, the sample behaves like an RGG on a locally Euclidean patch with well-controlled volume growth. Under these assumptions, the standard $k\asymp \log n$ and $r_n\asymp ((\log n)/n)^{1/d}$ connectivity thresholds still apply, with curvature and non-uniform sampling absorbed into the constants $\underline c,\overline c$. We have made these local mass assumptions explicit and to explain more clearly how the manifold model and Niyogi–Smale–Weinberger-type results justify the use of RGG asymptotics in our setting.

---

### Official Review · Reviewer_Zg8j · 2025-10-31

**Soundness:** 3
**Presentation:** 3
**Contribution:** 3
**Rating:** 6
**Confidence:** 2

**Summary:**

The paper introduces a theoretical and algorithmic framework for clustering based on the manifold hypothesis. It defines a geometric separability criterion using the ratio between inter-manifold offset and sampling fill distance, proving threshold theorems for when kNN graphs connect or separate manifold components. Building on this criterion, the authors propose a Manifold-Based Clustering algorithm that avoids manual density thresholds and provides a confidence bracket on the number of clusters. Experimental studies on synthetic, high-dimensional, and biological datasets demonstrate that MBC achieves accuracy comparable to state-of-the-art methods.

**Strengths:**

- The paper provides a rigorous analysis connecting geometric properties (reach, fill distance) with random graph connectivity, establishing explicit separation thresholds with proofs.
- The proposed MBC directly translates theory into a practical algorithm that is nearly parameter-free and interpretable through the offset/distance ratio, which is much more preferred in practice compared to density-based methods (DBSCAN, HDBSCAN) that require hand-tuned thresholds.
- Experimental results on different datasets show the effectiveness of the theory and the proposed method. Especially on the challenging data: the neural case study effectively demonstrates the method's utility in real scientific contexts, where MBC's brackets correctly distinguish between genuinely clustered and unclustered (V1) data while baselines force spurious partitions.

**Weaknesses:**

- There is limited scalability analysis in this paper. While computational complexity is briefly mentioned (O(n^2D) in high dimensions with brute-force k-NN), there's insufficient empirical evaluation of runtime scaling. The largest dataset appears to be MNIST (n~60k), leaving questions about performance on modern large-scale applications (for example, n > 1M)
- More baseline methods can be compared. In particular, while DBSCAN, OPTICS, and HDBSCAN are tested, the paper omits comparisons to spectral or graph neural clustering methods that could be more competitive. Besides, the comparison focuses primarily on density-based methods; other manifold-aware approaches (e.g., recent diffusion-based or topological clustering methods) are not evaluated.
- There is limited exploration of sensitivity. Although claimed parameter-light, practical sensitivity to \alpha, and the rescue-step constants (\theta, q, etc.) is not systematically analyzed.

**Questions:**

- Could the authors elaborate on the computational cost of the method (time and memory) relative to HDBSCAN or spectral clustering, especially in high dimensions?
- How sensitive is the reported confidence bracket to the choice of parameters?

---

> ### Author Response · Authors · 2025-11-23
>
> We thank the reviewer for the careful and constructive feedback. We are grateful for the positive evaluation of our geometric analysis connecting reach and fill distance to random graph connectivity, the explicit separation thresholds we establish, and the way MBC translates this theory into a practical, nearly parameter-free algorithm based on an interpretable offset–to–distance ratio. We also appreciate that the reviewer highlighted the neural case study as evidence of practical utility, especially the behavior of the bracket in distinguishing genuinely clustered from unclustered data where baseline methods force partitions.
>
> In the revised manuscript, we have incorporated a substantial set of improvements addressing the reviewer’s insights, as listed below:
>
> - “There is limited scalability analysis in this paper. While computational complexity is briefly mentioned ($O(n^2 D)$ in high dimensions with brute-force k-NN), there's insufficient empirical evaluation of runtime scaling. The largest dataset appears to be MNIST (n~60k), leaving questions about performance on modern large-scale applications (for example, $n$ > 1M)”
>
> We expanded the complexity discussion to state more explicitly that MBC’s dominant cost in building, particularly the bracket, is in building the k nearest neighbor graph for which fast algorithms exist. Using the optional dtm-rescue step increases the complexity, but for capturing uncertainty with the bracket additional computation on top of the knn computation remains linear.
>
> - “More baseline methods can be compared. In particular, while DBSCAN, OPTICS, and HDBSCAN are tested, the paper omits comparisons to spectral or graph neural clustering methods that could be more competitive. Besides, the comparison focuses primarily on density-based methods; other manifold-aware approaches (e.g., recent diffusion-based or topological clustering methods) are not evaluated.”
>
> Beyond the density-based methods in the main table, we have added spectral clustering and extended DBSCAN/OPTICS/HDBSCAN comparisons in the appendix, and for the neural data in particular we report parameter sweeps in the appendix and the limitations sections to highlight how sensitive these methods are to hyperparameter choices.
>
> - “There is limited exploration of sensitivity. Although claimed parameter-light, practical sensitivity to $\alpha$, and the rescue-step constants ($\theta$, $q$, etc.) is not systematically analyzed.”
>
> We strengthened the connection between the bracket and the theory by adding __Corollary 4.1__, which makes precise how the confidence window in degree, determined by alpha and A controls the behavior of the bracket. Thereby embracing the uncertainty of these parameters in a principled way to capture the underlying uncertainty of the main fill distance heuristic from the threshold.
>
> Below, we respond to the specific questions raised:
>
> - “Could the authors elaborate on the computational cost of the method (time and memory) relative to HDBSCAN or spectral clustering, especially in high dimensions?”
>
> In the revised algorithm and discussion sections, we now state that MBC and HDBSCAN share the same leading cost of kNN graph construction, with MBC replacing MST/hierarchy construction by local filtering passes, which are cheaper in both time and memory.  Spectral clustering uses the same neighborhood graph but additionally requires eigen-decomposition of a graph Laplacian, which is significantly more expensive in both runtime and memory footprint, especially as nnn grows.
>
> - “How sensitive is the reported confidence bracket to the choice of parameters?”
>
> We tie the bracket directly to the separation thresholds: for a given alpha and A, if the offset/fill distance ratio​ lies comfortably above the upper threshold (or below the lower one), then every degree in the window yields the same component count and the bracket collapses, while values of the ratio​ in the intermediate strip can produce a wide interval. This pattern is what we observe empirically: synthetic datasets with clear separation have tight brackets across parameter choices, whereas the retinal data retain wide brackets across settings, indicating that the uncertainty is inherent to the sampling and separability rather than solely a byproduct of parameter values.

---

### Official Review · Reviewer_opqQ · 2025-11-01

**Soundness:** 3
**Presentation:** 3
**Contribution:** 2
**Rating:** 4
**Confidence:** 3

**Summary:**

The paper addresses a fundamental question in unsupervised learning: under what conditions does data naturally form multiple clusters rather than a single connected structure?

Leveraging the manifold hypothesis, the authors model the data as samples drawn from a manifold which may consist of multiple disconnected components. In this framework, clustering becomes a topological decision problem—determining whether tghe manifold M is connected or separable by a measurable gap.

Using tools from random geometric graph theory, the paper derives sharp connectivity thresholds for k-nearest-neighbor (kNN) graphs: above a critical ratio Δ/h, components remain disconnected with high probability; below this threshold, cross-edges emerge and the structure becomes connected.

Building on this theoretical insight, the authors propose MBC, a parameter-efficient geometric clustering algorithm that exploits these connectivity properties for robust unsupervised partitioning.

**Strengths:**

The paper elegantly bridges topological geometry concepts—such as reach and fill distance—with graph-based clustering through explicit probabilistic threshold analysis.

Its theoretical framework unifies manifold geometry, random geometric graph theory, and noise analysis into a coherent and rigorous foundation for understanding cluster connectivity.

**Weaknesses:**

The theoretical results may have limited relevance in high-dimensional settings (d large), which could restrict the scalability of the approach for large or complex datasets.

Algorithm 1 involves several parameters that are fixed without clear justification—their selection appears somewhat ad hoc (see Step 1 of the algorithm). A discussion or sensitivity analysis of these parameter choices would strengthen the paper.

In addition, the presentation needs improvement. The TwoScaleDTM function in Step 14 is not a standard algorithm and should be explained more clearly. Although a definition is briefly provided in lines 251–253, the formula used in Step 14 differs from that description. Moreover, several components are undefined or ambiguous: what does CC represent in Step 19? what is GM in Step 20? and where does K in Step 20 come from? Finally, Section A.2 is currently empty and should either be completed or removed.

**Questions:**

Since the proposed algorithm is built upon the k-nearest-neighbor (kNN) framework, it would be reasonable to include a standard kNN-based clustering method as a baseline for comparison. This would help clarify the performance gains introduced by the proposed modifications.

The implication of Theorem 1 is also not entirely clear. It guarantees only that the “Euclidean geometric-mean gate followed by triangle support” introduces no cross-component edges, which is a relatively weak form of assurance. One would expect a stronger theoretical result—such as a bound on the clustering error rate or guarantees on correctly identifying the number of clusters—to better demonstrate the algorithm’s reliability and effectiveness.

---

> ### Author Response · Authors · 2025-11-23
>
> We thank the reviewer for the thoughtful and insightful feedback. We particularly appreciate the recognition that the paper aims to bridge geometric notions such as reach and fill distance with graph-based clustering via explicit probabilistic thresholds, and to unify manifold geometry, random geometric graph theory, and noise analysis into a coherent framework for understanding cluster connectivity. This perspective captures the core conceptual contribution we hoped to convey.
> In the revised manuscript, we have incorporated a substantial set of improvements addressing the reviewer’s insights,  as listed below:
>
> - “The theoretical results may have limited relevance in high-dimensional settings (d large), which could restrict the scalability of the approach for large or complex datasets.”
>
> We agree, the theoretical thresholds depend on the intrinsic dimension and local mass bounds rather than the ambient dimension, and in practice we work in a reduced representation with effective dimension given by PCA (90% variance, capped at 64), as described in the revised algorithm section. The high-dimensional Gaussian blob experiment (50D after PCA) illustrates that when separation is favorable at this intrinsic scale, MBC matches strong baselines, and we explicitly note in the limitations that extending the approach to learned embeddings is an important direction for genuinely high-dimensional applications.
>
> - “Algorithm 1 involves several parameters that are fixed without clear justification—their selection appears somewhat ad hoc (see Step 1 of the algorithm). A discussion or sensitivity analysis of these parameter choices would strengthen the paper.”
>
> We now distinguish more clearly between the theoretically motivated scale for k* which comes directly from the connectivity threshold, and the rescue constants which are fixed once and used unchanged across all datasets. We added brief sensitivity checks to the appendix and softened the claim to “parameter-light,” emphasizing that these constants are chosen for robustness rather than tuned per dataset.
>
> - “In addition, the presentation needs improvement. The TwoScaleDTM function in Step 14 is not a standard algorithm and should be explained more clearly....Moreover, several components are undefined or ambiguous: what does CC represent in Step 19? what is GM in Step 20?...”
>
> We apologize for this lapse and have rewritten the description of the two-scale DTM so that the definition in the main text matches the formula used in Algorithm 1. We refer to it consistently as a directional two-scale distance-to-measure statistic. Ambiguous labels such as “CC,” “GM,” and “K” have been replaced with explicit notation for connected components, the geometric-mean gate, and the bracket endpoints. We’ve added a more in-depth section on the algorithm in the appendix as well.
>
> Below, we respond to the specific questions raised:
>
> - “Since the proposed algorithm is built upon the k-nearest-neighbor (kNN) framework, it would be reasonable to include a standard kNN-based clustering method as a baseline for comparison. This would help clarify the performance gains introduced by the proposed modifications.”
>
> In response, we added a kNN connected-components baseline in the appendix using spectral clustering. However, we would like to emphasize that existing density methods such as DBSCAN do indeed use knn graphs, and more conventional approaches often assume the number of clusters is known i.e. spectral clustering. Our comparisons show that plain kNN components do not work well on the real world data that motivated our study.
>
> - “The implication of Theorem 1 is also not entirely clear. It guarantees only that the “Euclidean geometric-mean gate followed by triangle support” introduces no cross-component edges, which is a relatively weak form of assurance. One would expect a stronger theoretical result—such as a bound on the clustering error rate or guarantees on correctly identifying the number of clusters—to better demonstrate the algorithm’s reliability and effectiveness.”
>
> We appreciate this feedback! The main theoretical assurance described is accounted for by Theorem 3.3, which through our  revised Figure 1, clarifies that our theoretical results imply that, as a function of offset, sample complexity, and the underlying geometry of the data (reach, dimension, etc.) data can be identified as separable, non-separable or transitory in this probabilistic setting. Theorem 4.1 extends this inherent uncertainty to the tubular noise setting, and shows our algorithm performs no worse under strict assumptions on the geometric complexity of the data manifold. To provide a more theoretical justification for the inherent uncertainty captured by our algorithm,  we included Corollary 4.1 in our revision which depicts how our bracket carries the error of the threshold, based on the probabilistic estimate of the correct fill distance to provide a bound on the number of clusters present in the data.

---

### Official Review · Reviewer_kHFX · 2025-11-01

**Soundness:** 2
**Presentation:** 2
**Contribution:** 2
**Rating:** 2
**Confidence:** 3

**Summary:**

The paper introduces a new approach to clustering based on the manifold hypothesis, which assumes high-dimensional data lies on or near lower-dimensional manifolds. The core idea is that instead of forcing data into clusters, it is better to first understand whether distinct clusters exist at all at the available sampling scale. The clustering criterion is based on intuition similar to Gaussian Mixture Models—comparing separation between groups to dispersion within groups—but generalized to the case of Riemannian manifolds. Specifically, the authors propose comparing the offset Δ (minimal distance between manifold components) to the fill distance h (worst-case sampling gap), forming the ratio ρ = Δ/h that governs separability. When this ratio exceeds a threshold, manifold components remain disconnected in a k-nearest-neighbor graph; otherwise, they fuse together. The method is validated through experiments on both synthetic benchmarks and real-world neural data, demonstrating its ability to recover clusters when they are clearly separated and to report uncertainty (via a "bracket" on the number of components) when the data is ambiguous.

**Strengths:**

The main ideas of the method are interesting and promising: assuming clusters belong to different manifolds is reasonable and the kNN are usually a very good basis for clustering. The core concepts are rigorously stated and proof sketches are given.

1. Originality. The extension of the Gaussian Mixture Model framework to Riemannian manifolds is an elegant contribution and the core conceptual framework appears sound and well-motivated. The authors adapt the variance-to-separation intuition from GMMs to the manifold setting by replacing variance with fill distance and inter-cluster distance with manifold offset.

2. Quality. The paper provides theoretical rigor and empirical validation. Key claims are supported by formal proofs (Theorems 3.3, 4.1), however, the empirical evaluation is weak.

3. Significance. In my opinion, the contribution addresses an important gap in clustering methodology by reducing hyperparameter dependence (line 272). Unlike density-based methods that require manual scale thresholds, MBC provides a principled, geometry-driven decision criterion and reports uncertainty through a monotone bracket on the cluster count. This reduction in required prior knowledge makes the method more accessible and robust across application domains.

**Weaknesses:**

W1) Empirical evaluation: Unfortunately, the experimental results are not convincing. There are only few datasets tested. Most of them are low-dimensional toy datasets, where MBC works well, however, on all higher-dimensional datasets it yields NMI and ARI value of 0.0 -- This holds for Digits, MNIST, Retina (subset as well as full dataset). So I do not see where MBC would be used in practice.

Furthermore,  the parameter settings of competitive methods seem to have not been chosen well: in Table 1, OPTICS and HDBSCAN are reported to not find the correct clusterings for Two Moons and Concentric Circles, which both are datasets that are known to work well together with those algorithms.

W2) The competitive methods are only basic clustering methods that all work very similar. Please also compare to similar but different methods (e.g., Spectral clustering, Density Peaks), and newer methods and improvements on DBSCAN (e.g., [0,1,2])

[0] Chazal, F., Guibas, L. J., Oudot, S. Y., & Skraba, P. (2013). Persistence-based clustering in Riemannian manifolds. Journal of the ACM (JACM), 60(6), 1-38.

[1] Xu, X., Ju, Y., Liang, Y., & He, P. (2015, October). Manifold density peaks clustering algorithm. In 2015 Third international conference on advanced cloud and big data (pp. 311-318). IEEE.

[2] Deng, C., Zhang, Q., Zhou, X., Zhang, S., Wang, G., & Xu, W. (2025). Density peaks clustering algorithm integrating manifold distance and mutual nearest neighbors. Pattern Recognition, 112554.


W3) The paper is quite hard to follow and could benefit from figures guiding the readers.

**Questions:**

Q1) Why were baseline methods used mostly with default settings and were any tuned versions tried?

Q2) How complex can the data get before no meaningful clustering is possible anymore with your method ? What is the expected runtime for each data complexity factor?

Q3) Why did you not compare to newer methods?

---

> ### Author Response · Authors · 2025-11-23
>
> We thank the reviewer for the thoughtful, encouraging, and highly constructive feedback. We especially appreciate the positive assessment of our main ideas: extending the variance–to–separation intuition from Gaussian mixture models to a manifold setting via fill distance and offset, formulating the core concepts rigorously and supporting them with formal results. We are also grateful that the reviewer highlighted the value of a principled, geometry-driven decision criterion and the role of the monotone bracket in reducing hyperparameter dependence.
>
> In the revised manuscript, we have incorporated a substantial set of improvements addressing the reviewer’s insights, as listed below:
>
> - __(W1)__ “Empirical evaluation: Unfortunately, the experimental results are not convincing. There are only few datasets tested... So I do not see where MBC would be used in practice. Furthermore, the parameter settings of competitive methods seem to have not been chosen well...”
>
> We have clarified that MBC is intended as a geometric test of clusterability and scale rather than a universal replacement for existing methods: when the offset–to–fill-distance ratio is large (Moons, Circles, Gaussian blobs), MBC matches strong baselines, while in entangled or under-sampled regimes (Digits, MNIST, Retina) it intentionally widens the bracket instead of forcing a partition. The revised neural case study now shows that on V1 MBC returns a single component with a narrow bracket [1,3] consistent with physiological expectations, whereas on Retina it yields a wide bracket containing the known number of clusters and nontrivial agreement with labels at its upper end. We also expanded the empirical section and appendix to include tuned runs of HDBSCAN/OPTICS/Birch/DBSCAN and note that, as expected, these baselines recover the correct clusters on Moons and Circles when appropriately tuned.
>
> - __W2)__ “The competitive methods are only basic clustering methods that all work very similar. Please also compare to similar but different methods (e.g., Spectral clustering, Density Peaks), and newer methods and improvements on DBSCAN (e.g., [0,1,2])”
>
> We broadened the comparison to include additional clustering methods for the neural data in particular, with varying choices of hyperparameters. The main table focuses on HDBSCAN and BIRCH for readability, while the extended baselines and their hyperparameter sensitivity are presented in the appendix.
>
> - __W3)__ “The paper is quite hard to follow and could benefit from figures guiding the readers.”
>
> To improve readability, we revised our Fig. 1 to illustrate the separable/transitory/nonseparable regimes and neural examples more explicitly using figures from the appendix, and we added Fig. 2 to offer a more clear outline of the MBC pipeline. We further revised our introduction and included an additional corollary explaining the bracket and its ability to capture the uncertainty in the threshold.
>
> Below, we respond to the specific questions raised:
>
> - __Q1)__ “Why were baseline methods used mostly with default settings and were any tuned versions tried?”
>
> In the main experiments we used library defaults to avoid cherry-picking hyperparameters, and we now report in the appendix tuned runs and parameter sweeps for DBSCAN, OPTICS, HDBSCAN, and spectral clustering. The reason was because we intended to focus on the application of these unsupervised clustering algorithms for exploratory data analysis, where a priori the desired number of clusters, and therefore the hyperparameters are unknown.
>
> - __Q2)__ “How complex can the data get before no meaningful clustering is possible anymore with your method ? What is the expected runtime for each data complexity factor?”
>
> The complexity of the data is a function of the complexity of the manifold (i.e. the dimension, reach/curvature, etc.) as well as the number of points, and these are estimable directly from the constants in the threshold theorem. Informally, if the ground truth manifold is highly curved more points and a greater offset are required, and this bound on the sample complexity increases exponentially with dimension. Importantly, our manifold-motivated approach at least makes such considerations part of the formulation.
>
> The fill-distance approximation captures the number of points, and the constants capture the manifold geometry. These don't use the runtime of the algorithmic approach, because we adopt a heuristic of taking these often unestimatable constants, so we use a reasonable approximation under simplifying assumptions to make the heuristic for separability tractable to compute.
>
> - __Q3)__ “Why did you not compare to newer methods?”
>
> The purpose of focusing on these mainstream methods is because of their prevalence in existing biological literature, and our goal was to emphasize the inherent overfitting that these methods impose on the data by not accounting for the uncertainty inherent in cluster separability.

---

> > ### Comment · Reviewer_kHFX · 2025-11-26
> >
> > Dear authors,
> > thanks for putting in time and effort into the answer and the revision.
> > Unfortunately, it is very hard to check the actual changes as you neither highlight them in the paper nor refer to any section that has been changed with the actual number of the section. I'd greatly appreciate it (and I guess the other reviewers, too) if you mark and describe clearly where the changes were done.
> >
> > Other notes:
> > - There is an empty Section A.3 (A.2 in the original paper).
> > - What value does it have to show k-Means and the version of spectral clustering you are using in Table 5? They will always find the number of clusters given by the user k.

---

> > > ### Author Response · Authors · 2025-11-26
> > >
> > > Thank you for your quick reply. We have resubmitted an updated version of the paper with all main changes highlighted. Captions are also highlighted to indicate revisions made to the figures in the main text. In response to your comments:
> > >
> > > - We have removed the empty section.
> > >
> > > - We agree! In the appendix table, we had not clearly stated that we varied the number of clusters, $k$, and reported the resulting clusters after treating very small clusters (i.e., those with less than 2% of the data points) as noise. Our intention was to simulate the practical scenario where the true number of clusters is unknown, so we expect practitioners to vary the $k$ hyperparameter. We have now clarified this point in the table caption.
> > >
> > > Please let us know if there are any additional questions or if further clarification would be helpful.

---

### Author Response · Authors · 2025-11-23
**General response to all reviewers**

We thank all four reviewers for their time, careful reading, and detailed feedback. We appreciate in particular that several of them highlighted the conceptual appeal of a geometric, nearly parameter-light clustering criterion, and the attempt to formalize aspects of the manifold hypothesis into an operational decision rule.

After reading the reviews, it became clear to us that two central aspects of our contribution were not sufficiently visible in the original draft:
- the bracket as a primary objective—how it is constructed, how it connects back to the separation theorem, and what kind of “uncertainty” it is meant to represent; and
- how this bracket behaves on complex real-world datasets, especially the retina and V1 recordings that motivated the work.
In the revision, we substantially edited the Introduction, Background, and Algorithm sections, reworked Figures 1 and 2, and added a corollary explicitly tying the bracket to the geometric thresholds.

We provide the summarized main changes below.

- __Clarified the problem framing and made the bracket central in the introduction__
 (addresses __kHFX__, __Zg8j__ and __gyYv__)

We rewrote the introduction to emphasize that our goal is not to outperform existing algorithms on any given benchmark, but to __provide a geometric test of clusterability and sampling scale, with the bracket as the main object__. The new opening explicitly contrasts artificial toy separations with the neuroscience problem and motivates the bracket:
_“We embrace the uncertainty directly, and propose an algorithm that yields a bracket [1–9] for the number of clusters. The algorithm is based on a topological analysis, and takes both shape and sampling into consideration. It follows that clusters may be easily separable, nonseparable, or the data may lie in a transitory domain.”_

- __Reorganized Theorem 1 and added a corollary that pushes uncertainty in the scale down to the separation threshold__ (__opqQ__, __Zg8j__ and __gyYv__ )

We restructured Theorem 1 so that assumptions appear as labeled subheads (“Local mass bounds,” “Sampling,” “Graph construction”) and the core “Threshold statement” focuses on what happens when $\Delta/h_{\max}$ lies above or below the critical constants.
Further, to connect this more directly to the bracket, we added an explicit “Uncertainty Bracket” paragraph and Corollary 4.1 which explicitly show how the degree window $[k_{\mathrm{low}},k_{\mathrm{high}}]$ translates into a controlled uncertainty band on $\Delta/h_{\max}$

The corollary and its discussion make clear that the bracket is not purely heuristic: it is designed to “push’’ the probabilistic uncertainty in $k^\star$ (due to sampling fluctuations) down to the scale of the separation threshold, and that the relative width $(k_{\mathrm{high}} - k_{\mathrm{low}})/k^\star$ remains $O(A^{-1/2})$.

- __Clarified the algorithm, parameter choices, and “parameter-light” claim; cleaned up notation__  (__opqQ__, __Zg8j__ and__gyYv__)

In the MBC Section and Algorithm pseudocode, we now clearly separate the theoretically motivated $k^\star=\lceil A\log(4n/\delta)\rceil$ from the fixed rescue constants $(\theta,q_\tau,c,|S_i|)$, and we explicitly softened the language surrounding a notion of “hyperparameter-free”, emphasizing that __the choices of hyperparameter are not pertinent to the point of our described MBC algorithm__. Ambiguous labels such as “CC,” “GM,” and “K” have been revised.

- __Reframed the role of synthetic experiments and expanded baselines, including tuned methods and kNN-based baselines__ (__kHFX__, __Zg8j__,__opqQ__ and __gyYv__)

We now include the __low-dimensional synthetic examples explicitly as sanity checks__ for the offset-fill-distance picture in the appendix and __focus the narrative on how the bracket behaves as $\Delta/h$ varies__:
_“Our results align with the offset-fill-distance picture: when $\Delta/h$ is large (Moons, Circles), MBC recovers ground truth with narrow brackets; on high-D separated blobs, MBC matches OPTICS/HDBSCAN; when embeddings are entangled (Fashion-MNIST, MNIST) …”_

The main table now focuses on HDBSCAN and BIRCH for readability (Table 1), but Appendix tables add the __results for other common baselines (DBSCAN, OPTICS, spectral clustering, etc.)__, together with results for parameter sweeps, particularly on the neural datasets.

Taken together, these changes aim to clarify that __MBC is a geometric, scale-calibrated diagnostic of clusterability rather than a one-size-fits-all replacement for existing clustering pipelines__. The bracket is now explicitly tied to the underlying separation theorem and is presented as the primary output, especially in ambiguous real-world regimes like the retina and V1 data where our goal is to surface—and not conceal—the uncertainty imposed by sampling and representation.

We look forward to your responses, and are happy to answer any additional questions. Thank you for your valuable time.

---

### Meta-Review · Area_Chair_Tp51 · 2025-12-28

**Summary:**

This paper addresses the clusterability problem based on the manifold hypothesis. The authors formulate the clustering problem to a topological decision problem for determining when the manifold is connected or has disconnected components. Based on random geometric graph theory, the paper derives a critical threshold for the kNN graph in terms of the offset-to-fill-distance ratio. With this theoretical insight, the paper proposes a manifold-based clustering (MBC) algorithm and provides a confidence bracket on the number of clusters. Experimental studies on synthetic and real data show that MBC achieves comparable performance to SOTA methods.

The reviewers raised several common weakness and questions:

- scalability in terms of high dimension and sample size;
- insufficient comparison with baseline methods;
- over-simplified setting, limited data size and sensitivity exploration in numerical experiments.

**Reviewer Concerns:**

In response, the authors revised the paper by emphasizing their main contribution is not to beat the SOTA performance, but rather to provide a geometrically aware test of clusterability with the bracket as the main objective. The authors also clarified the algorithm and parameter choices. Main Theorem 1 was structured accordingly.

Reading through the discussions, I appreciate that the rebuttal partially addressed the reviewers’ concerns. However, I still share some concerns as the reviewers on the practical applicability of MCB as a clustering tool. Scalability is not clear (for high-dimensional data of modern large size), where other computationally efficient clustering methods do exist. The theoretical insight for manifold components testing largely relies on the random geometric graph limits, which might be too narrow in scope.

**Reviewer Scores:**

4 reviewers submitted their comments and scores (2/4/2/6) with confidence (3/3/4/2), with average score 3.5 and average confidence 3.

---

### Decision · Program_Chairs · 2026-01-26

Reject